# Mitigating Perceptual Judgment Bias in Multimodal LLM-as-a-Judge via Perceptual Perturbation and Reward Modeling

**Seojeong Park** [* 1]  **Jiho Choi** [* 1]  **Junyong Kang** [1]  **Seonho Lee** [2]  **Jaeyo Shin** [1]  **Hyunjung Shim** [1]

## Abstract

Recent multimodal large language models have demonstrated strong reasoning ability, yet their reliability as automated evaluators remains limited by a critical weakness: when visual evidence conflicts with textual cues, MLLM judges tend to reward plausible narratives over perceptually correct answers. We identify and systematically analyze this phenomenon, which we term *Perceptual Judgment Bias*. Through controlled visual perturbations, existing multimodal judges frequently anchor on the response text instead of their own visual perception, leading to inconsistent and non-verifiable evaluations. To address this issue, we introduce the *Perceptually Perturbed Judgment Dataset*, which constructs minimally edited counterfactual responses that isolate perceptual errors and enable verifiable supervision. Building on this dataset, we develop a unified training framework that optimizes a verifiable batch-ranking reward with GRPO, achieving coherent global ordering without explicit pairwise labels. Experiments across diverse MLLM-as-a-Judge benchmarks show that our approach substantially improves perceptual fidelity, ranking coherence, and alignment with human evaluation. Our method establishes a principled and scalable paradigm for training multimodal judges that are perceptually grounded, interpretable, and robust to visual–reasoning conflicts. Project Page: https://perception-judge.github.io/

## 1. Introduction

Multimodal large language models (MLLMs) are rapidly reshaping the landscape of vision–language reasoning, grounding, and understanding (Achiam et al., 2023; Hurst et al., 2024; Liu et al., 2023; Bai et al., 2023; Yang et al., 2025; Wang et al., 2025a). Their growing ability to integrate textual and visual information has led to impressive progress in instruction following and open-ended multimodal generation. Nevertheless, evaluating the quality of multimodal responses remains a fundamental challenge (Li et al., 2024; Chiang et al., 2024; Li et al., 2025a; Zhang et al., 2025). As MLLMs are increasingly used in decision-critical applications (Wang et al., 2025c; Li et al., 2024; Pu et al., 2025; Li et al., 2023), reliable evaluation becomes essential not only for fair comparison among models but also for scalable replacement of costly human evaluation as model sizes and benchmark scope expand. Beyond passive assessments, a strong evaluation framework can actively contribute to iterative model improvement by generating reward signals that guide MLLM alignment and optimization (Yuan et al., 2024; Li et al., 2025b; Ouyang et al., 2022; Zheng et al., 2023; Sun et al., 2023).

The gold standard for such evaluation has traditionally been human judgment (Ouyang et al., 2022; Dai et al., 2023; Yuan et al., 2023); however, this approach is prohibitively costly, time-consuming, and often suffers from inter-annotator inconsistency. These scalability constraints have catalyzed the emergence of the *LLM-as-a-Judge* paradigm (Chiang et al., 2024; Li et al., 2024; Zhu et al., 2023), which utilizes language models as **automated evaluators** to provide structured scores and interpretable justifications. Recent advancements have further extended this to the multimodal domain through *MLLM-as-a-Judge* (Chen et al., 2024; Pu et al., 2025; Xiong et al., 2025; Ko et al., 2025; Waheed et al., 2025; Kim et al., 2024; Zhang et al., 2025), enabling reasoning-aware assessment of vision-language responses at scale.

Despite their potential, existing MLLM judges (Chen et al., 2024; Xiong et al., 2025; Ko et al., 2025) often produce linguistically plausible yet visually ungrounded evaluations, a phenomenon we formalize as *Perceptual Judgment Bias* (Section 3.2). This bias represents a systematic deviation from human judgment rooted in two distinct failure modes: **(a) insufficient perceptual capability**, where the judge over-rewards logically coherent but visually incorrect re-

---

[*]Equal contribution  [1]Graduate School of Artificial Intelligence, KAIST, Republic of Korea [2]KRAFTON. Correspondence to: Hyunjung Shim <kateshim@kaist.ac.kr>.

*Proceedings of the 43rd International Conference on Machine Learning*, Seoul, South Korea. PMLR 306, 2026. Copyright 2026 by the author(s).

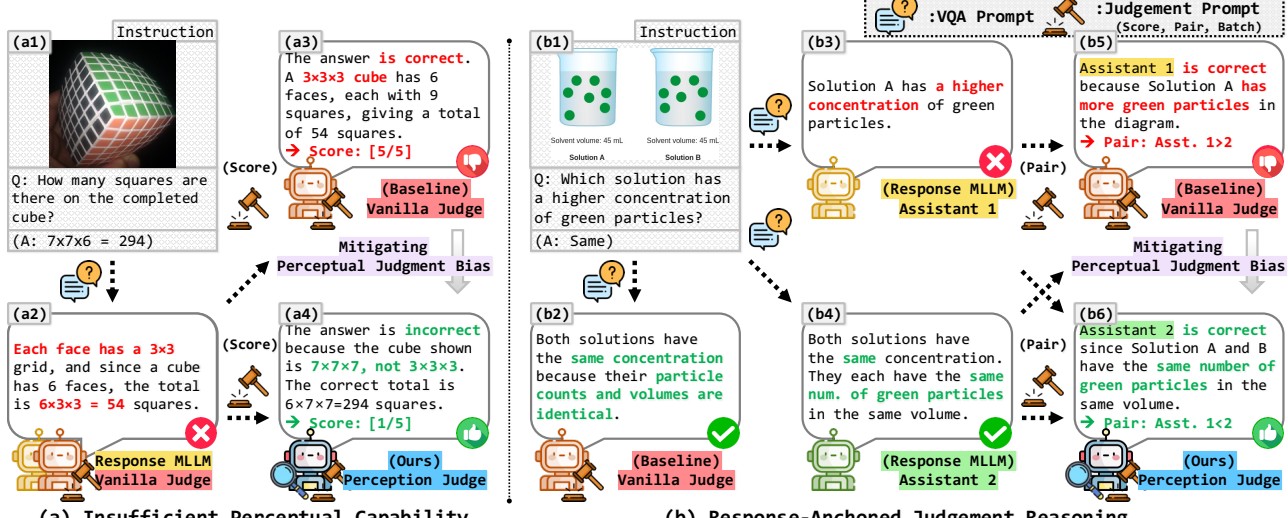

*Figure 1.* **Perceptual judgment bias in MLLM judges.** (a) When perceptual capability is insufficient, a judge may produce incorrect visual descriptions (a2) and assign high scores (a3) to perceptually wrong responses (a2). (b) Even when the judge's own perception aligns with humans (b2), it may still prefer (b5) visually inconsistent responses (b3) compared to the response with correct perception (b4). We introduce **Perception-Judge**, an MLLM judge trained with reinforcement learning on a systematically designed perception-grounded dataset, **PPJD**, which effectively mitigates these perceptual biases in MLLM judgment (a4),(b6).

sponses due to inaccurate visual understanding (Figure 1 a), and **(b) response anchoring**, where the judge fails to ground its decision in its own perception, instead defaulting to visual descriptions provided within the response text (Figure 1 b). Together, these behaviors reveal a fundamental misalignment between perception and reasoning, suggesting that current evaluators often prioritize linguistic consistency over direct visual evidence. The formalization of this bias underscores the necessity for evaluation frameworks that explicitly enforce perception-based verification as a prerequisite for reasoning, thereby motivating our proposed perception-guided paradigm.

To address these limitations, we propose *Perception-Judge*, a multimodal evaluator that reinforces perceptual grounding in model assessment. We first quantify *Perceptual Judgment Bias* in Section 3.2 by decomposing baseline judgment errors into two failure modes: insufficient perceptual capability and response anchoring. As quantified in Table 1, baseline judges incur substantial overall error rates exceeding 20%, indicating that both failure modes degrade judgment performance. A more detailed analysis with controlled visual perturbations further reveals that baseline judges can reliably identify responses with compound errors, achieving high accuracy above 0.8. However, when the error is isolated to perceptual inconsistency alone, their performance drops substantially by over 10% (Figure 2). These observations motivate an evaluator that explicitly enforces perceptual verification as a prerequisite for reliable judgment. To mitigate this bias, we construct the *Perceptually Perturbed*

*Judgment Dataset (PPJD)* in Section 3.3. PPJD provides a specialized training foundation by injecting controlled visual inconsistencies into otherwise correct responses, producing quadruplets $(x_i, r_c, r_{r_p}, r_{r_{p+r}})$ that explicitly disentangle perceptual failures from reasoning errors. Unlike generic preference datasets, PPJD provides fine-grained, verifiable supervision that exposes subtle perceptual failures, serving as a robust foundation for perception-aware alignment. Using PPJD, we design a structured reward objective in Section 3.4 that requires judges to distinguish fully correct responses from those exhibiting perceptual errors, while further differentiating cases where perceptual and reasoning errors jointly occur through an explicit graded ordering. By enforcing perceptual verification as a prerequisite for high rewards, this design prevents judges from over-rewarding responses based on reasoning plausibility or response-described cues when visual grounding is incorrect.

This framework thereby achieves perception-aware evaluation and effectively mitigates the bias observed in prior models. Empirically, our approach delivers consistent performance gains across evaluation protocols. It improves batch-level metrics by up to 11% on Qwen3-VL-4B-Thinking and increases single-score prediction accuracy by 15% on Flex-Judge-7B. It also matches the performance of leading proprietary evaluators in score-based settings and surpasses many of them in batch-level consistency. This work establishes a principled framework for training evaluators under verifiable but non-differentiable supervision, enabling reliable and perception-aware multimodal judgment at scale.

## 2. Background & Preliminary

**MLLM-as-a-Judge & Visual Reasoning.** Prior work has extensively explored the *LLM-as-a-Judge* paradigm, where language models evaluate the quality of model-generated responses (Chiang et al., 2024; Li et al., 2024; 2025a; Thakur et al., 2025; Whitehouse et al., 2025; Saha et al., 2025; Chen et al., 2025a; Zhu et al., 2023). Recent studies further incorporate reinforcement learning (RL) and verifiable reward schemes to improve scoring reliability and preference consistency (Whitehouse et al., 2025; Chen et al., 2025a; Su et al., 2025; Wen et al., 2025; Yuan et al., 2024). This paradigm has been extended to multimodal settings as *MLLM-as-a-Judge*, enabling vision-language evaluation, but existing approaches mainly rely on supervised fine-tuning and often suffer from visual perception judgment bias (Chen et al., 2024; Ko et al., 2025; Pu et al., 2025; Waheed et al., 2025; Lee et al., 2024; Xiong et al., 2025). In parallel, recent works show that MLLMs frequently struggle with visual perception and grounding, and propose training-free methods, RL-based optimization, and data-centric strategies to improve visual reasoning (Chen et al., 2025b; Jian et al., 2025; Tian et al., 2025; Wang et al., 2025b; Chen et al., 2025c; Bigverdi et al., 2025). A more comprehensive review is provided in the appendix Section B.

**Group Relative Policy Optimization (GRPO).** Reinforcement learning has become a key approach for aligning (M)LLMs with human feedback through verifiable or preference-based rewards (Ouyang et al., 2022; Zheng et al., 2023; Yuan et al., 2023; Dai et al., 2023). However, traditional methods such as PPO (Schulman et al., 2017) require an explicit value network, often leading to instability under sparse or noisy rewards. GRPO (Shao et al., 2024; DeepSeek-AI et al., 2025) mitigates this issue by estimating *relative* advantages from multiple responses within a group, enabling stable and value-free optimization.

Given a prompt $x$ from the dataset $\mathcal{D}$, a policy $\pi_{\theta_{\mathrm{old}}}$ generates $n$ candidate responses $O = \{o_i\}_{i=1}^n$ with corresponding rewards $\mathcal{R} = \{R(o_i)\}_{i=1}^n$. GRPO normalizes rewards within each group to compute the relative advantage as: $\hat{\mathcal{A}}_i = \frac{R(o_i) - \mu(\mathcal{R})}{\sigma(\mathcal{R})}$, where $\mu(\mathcal{R})$ and $\sigma(\mathcal{R})$ denote the mean and standard deviation of group rewards $\mathcal{R}$ respectively. This intra-group normalization ensures that policy updates (Sutton et al., 1999) are invariant to reward magnitude and robust to inter-task reward variance, effectively stabilizing the optimization process (Li et al., 2025b; Xue et al., 2025).

The policy $\pi_\theta$ is updated to increase the likelihood of higher-quality responses while maintaining closeness to the behavior and reference policies $\pi_{\mathrm{ref}}$. The GRPO objective extends PPO's clipped surrogate loss with group-normalized advantages and KL regularization $\mathbb{D}_{\mathrm{KL}}$ as:

$$\mathcal{J}_{\mathrm{GRPO}}(\theta) = \mathbb{E}_{x \sim \mathcal{D}, \{o_i\}_{i=1}^n \sim \pi_{\theta_{\mathrm{old}}}(O|x)} \Big[$$
$$\frac{1}{n} \sum_{i=1}^n \min(r_i \hat{\mathcal{A}}_i, \mathtt{clip}(r_i, 1-\epsilon, 1+\epsilon)\hat{\mathcal{A}}_i) \quad (1)$$
$$- \beta \mathbb{D}_{\mathrm{KL}}(\pi_\theta \,\|\, \pi_{\mathrm{ref}}) \Big],$$

where $r_i = \frac{\pi_\theta(o_i|x)}{\pi_{\theta_{\mathrm{old}}}(o_i|x)}$ and $\beta$ controls regularization.

## 3. Proposed Method

We aim to improve the reliability of multimodal evaluation by addressing the perceptual judgment bias observed in multimodal LLM judges. We first define the *MLLM-as-a-Judge* task to formalize the multimodal evaluation setting (Section 3.1). Next, we identify the *Perceptual Judgment Bias*, where the judge prioritizes response-described cues or reasoning plausibility over their own visual perception. This bias is empirically analyzed through controlled case studies that compare human and model decisions under visual perturbations (Section 3.2). To mitigate this issue, we construct the *Perceptually Perturbed Judgment Dataset* (PPJD) (Section 3.3). The dataset utilizes a perception-aware judgment design that minimally alters visually grounded attributes while preserving the original reasoning, enabling counterfactual responses that expose perception errors. Building on this foundation, we develop a GRPO-based verifiable reward modeling framework, *Perception Judge*, which optimizes the judge for perceptual alignment with human evaluations (Section 3.4).

### 3.1. Problem Definition: MLLM-as-a-Judge

The MLLM-as-a-Judge task (Chen et al., 2024; Pu et al., 2025; Waheed et al., 2025) aims to evaluate the quality of $K$ multimodal responses $\{r_1, r_2, \ldots, r_K\}$ generated from a given input $x_i = \{q_i, v_i\}$. The textual component $q_i$ denotes an instruction (e.g., a question or a prompt), while $v_i$ represents the corresponding visual input (e.g., an image). A set of response models $\{\mathcal{M}_k\}_{k=1}^K$ produces candidate responses $\{r_k = \mathcal{M}_k(x_i)\}_{k=1}^K$, and the judge model $\mathcal{M}_{\mathrm{Judge}}$ evaluates how well each response aligns with the multimodal context. For each response $r_k$, the judge assigns a scalar judgment score $s$ as follows:

$$s_{(x_i, r_k)} = \mathcal{M}_{\mathrm{Judge}}(x_i, r_k), \quad k = 1, 2, \ldots, K. \quad (2)$$

Depending on the evaluation protocol, the judgment output can take one of three forms: (i) a scalar quality score $s_{(x_i, r_k)} \in [1, 5]$ for *score-based* evaluation, (ii) a preference relation between two responses $(r_a \succ r_b)$ for *pairwise* comparison, or (iii) an ordered sequence of candidates $(r_{\boldsymbol{\pi}_i(1)} \succ r_{\boldsymbol{\pi}_i(2)} \succ \cdots \succ r_{\boldsymbol{\pi}_i(K)})$ for *ranking-based* evaluation, where $\boldsymbol{\pi}_i$ denotes the permutation sorted by descending judgment scores $\{\mathcal{M}_{\mathrm{Judge}}(x_i, r_k)\}_{k=1}^K$. The goal of all

*Table 1.* **Analysis of perceptual judgment bias in MLLM judges.** Accuracy and error rates are decomposed into two failure modes: (a) insufficient perceptual capability and (b) response-anchored judgment reasoning. Perception-Judge-Flex-7B achieves higher accuracy and lower error rates than baseline judges, indicating improved perceptual grounding in multimodal evaluation.

| Model | Acc. ↑ | Error ↓ | | |
| --- | --- | --- | --- | --- |
| | | **Mode (a)** | **Mode (b)** | **Overall** |
| Qwen2.5-VL-7B | 69.5% | 14.0% | 16.4% | 30.5% |
| Flex-Judge-VL-7B | 76.6% | 9.4% | 14.1% | 23.5% |
| Perception-Judge-Flex-7B (Ours) | 85.7% | 6.7% | 7.6% | 14.3% |

evaluation protocols is to quantify how closely the judge's decisions align with human judgments, reflecting both perceptual grounding and reasoning consistency.

### 3.2. Perceptual Judgment Bias

Ideally, a multimodal judge should evaluate a response based on both its *correctness* and the judge's *own perception* of the input image $v_i$. In practice, however, MLLM judges systematically fail to penalize responses whose visual claims contradict the image, either because they misperceive the image themselves or because they fail to flag the false visual claims even when their own perception is correct. We term this phenomenon *Perceptual Judgment Bias*. Formally, let $\pi^\star(v_i)$ denote the human-verified visual facts, $\pi_{\text{Judge}}(v_i)$ the judge's own perception, $\pi_r(v_i)$ the visual content described in a response $r$, and $s_{(x_i,r)}$ its score for response $r$; a perceptual judgment error occurs when a visually inconsistent response $r_r$ is not penalized relative to the correct response $r_c$, i.e., $s_{(x_i,r_r)} \geq s_{(x_i,r_c)}$. As illustrated in Figure 1, this bias subsumes two failure modes distinguished by whether the judge's own perception is correct. In **Mode (a), insufficient perceptual capability**, the judge's perception is wrong, and it over-rewards a visually incorrect response,

$$s_{(x_i,r_r)} \geq s_{(x_i,r_c)},$$
$$\text{where} \quad \pi_{\text{Judge}}(v_i) \neq \pi^\star(v_i), \quad \pi_{r_r}(v_i) \neq \pi^\star(v_i); \quad (3)$$

whereas in **Mode (b), response-anchored judgment reasoning**, the judge perceives the image correctly in direct queries, but fails to apply this perception during evaluation, anchoring on the linguistically plausible response instead,

$$s_{(x_i,r_r)} \geq s_{(x_i,r_c)},$$
$$\text{where} \quad \pi_{\text{Judge}}(v_i) = \pi^\star(v_i), \quad \pi_{r_r}(v_i) \neq \pi^\star(v_i). \quad (4)$$

While related biases have been observed in MLLMs (Vo et al., 2025; Xia et al., 2025), we isolate and quantify this behavior specifically in the multimodal judgment.

To quantify the two modes, we construct a controlled perceptual-only comparison between $r_c$ and a perception-perturbed response $r_{r_p}$ on the MLLM-as-a-Judge benchmark (Chen et al., 2024). Using direct Visual Question Answering (VQA) correctness as a proxy for $\pi_{\text{Judge}}(v_i)$, we attribute an error to Mode (a) when the judge also fails

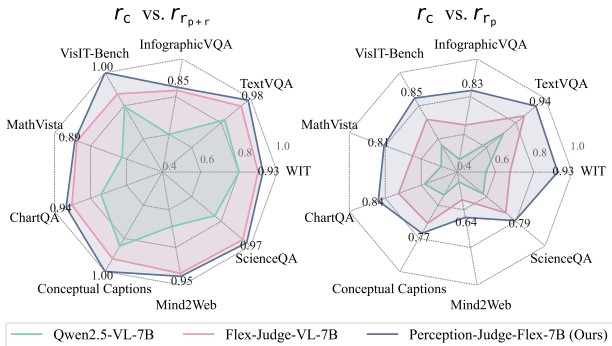

*Figure 2.* (Left) Accuracy of distinguishing the corrected response $r_c$ from a response with both perceptual and reasoning perturbations ($r_{r_{p+r}}$). Baseline MLLM maintains high accuracy. (Right) Accuracy of distinguishing $r_c$ from a response with only perceptual perturbations ($r_{r_p}$). Accuracy consistently drops across models, revealing their sensitivity to perceptual inconsistencies. Our Perception-Judge-Flex-7B improves performance in both settings and shows the largest gains in the perceptual-only condition, indicating a substantial reduction of perceptual judgment bias.

the VQA probe, and to Mode (b) when it answers correctly yet still misranks the responses. As reported in Table 1, both baselines incur substantial overall errors (30.5% and 23.5%), with Mode (b) comparable to or larger than Mode (a). This indicates the bias is not merely a recognition failure, but also a failure to couple recovered visual evidence with evaluative reasoning.

Beyond this mode decomposition, we additionally analyze how reliably a judge distinguishes the correct response $r_c$ from a perturbed one, depending on the type of perturbation, as shown in Figure 2. When the response is perturbed in both perception and reasoning ($r_{r_{p+r}}$), baselines distinguish it from $r_c$ with high accuracy, since the reasoning degradation supplies non-visual cues for rejecting a poor response. In contrast, when only the perception is perturbed while the reasoning remains plausible ($r_{r_p}$), their accuracy drops substantially. This shows that existing judges reliably reject a response only when its textual reasoning is also corrupted, and that strong performance on compound errors does not imply genuine perceptual grounding.

Taken together, these analyses motivate our perception-guided framework. Since baseline judges both anchor on textual cues (Mode (b)) and fail to penalize visually wrong yet reasoning-plausible responses ($r_{r_p}$), a reliable evaluator must learn to discriminate responses by their degree of perceptual corruption rather than their reasoning fluency. We therefore train the judge to internalize a graded ordering $r_c \succ r_{r_p} \succ r_{r_{p+r}}$, which forces it to distinguish isolated perceptual errors ($r_c \succ r_{r_p}$) from fully compounded failures ($r_{r_p} \succ r_{r_{p+r}}$). Enforcing this explicit global ordering effectively calibrates the judge against both failure modes, providing the rationale for the data construction (Section 3.3) and the batch-ranking reward (Section 3.4) described next.

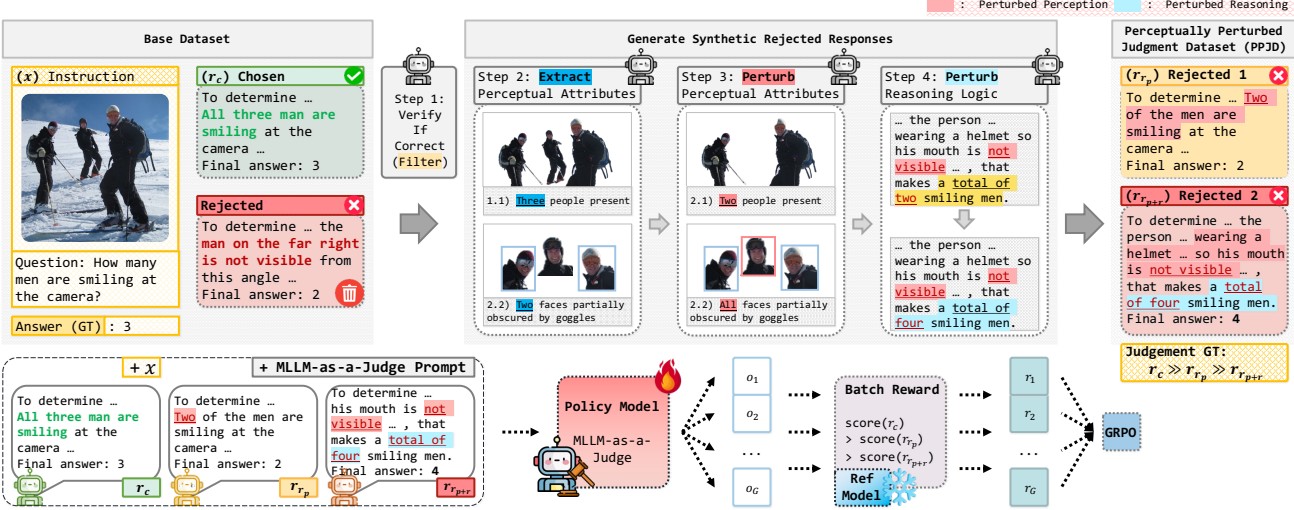

*Figure 3.* **Pipeline Overview.** Using perceptual perturbations, we construct the Perceptually Perturbed Judgment Dataset (PPJD). For each correct response, PPJD generates two perturbed variants. The first response, $r_{r_p}$, is produced by altering visually grounded attributes while preserving the original structure. The second response, $r_{r_{p+r}}$, is created by additionally degrading the original reasoning, resulting in a fully corrupted answer. We define the preference order of judgments as $r_c \succ r_{r_p} \succ r_{r_{p+r}}$. Utilizing these ordered triplets, we train the model with GRPO, where the reward is computed using the Levenshtein distance (Yujian & Bo, 2007).

## 3.3. Data Generation via Perceptual Perturbation

To investigate and mitigate perceptual judgment bias in multimodal reasoning, we construct the **Perceptually Perturbed Judgment Dataset (PPJD)**, which explicitly disentangles perceptual and reasoning inconsistencies. Grounded in the principle that reliable judgment should vary with perceptual differences, PPJD enables systematic analysis of how models incorporate visual evidence in their evaluations. Starting from the multimodal preference dataset MMPR (Wang et al., 2024), which provides chosen and rejected responses $r_c$ and $r_r$ respectively for each input $(x_i = \{q_i, v_i\})$, we retain $r_c$ as the perceptually and logically consistent reference.

We then generate a visually inconsistent but reasoning-preserved response $r_{r_p}$ by applying a controlled *perceptual perturbation* (p) to $r_c$ using a generative model (e.g., GPT (Achiam et al., 2023), Qwen (Bai et al., 2023; Yang et al., 2025)) or a pretrained judge model $\mathcal{M}_{\text{Judge}}$. Specifically, the perturbation modifies visually grounded attributes (e.g., object color, count, or spatial relations) mentioned in the original response, which are commonly analyzed in prior studies on multimodal perception errors (Vo et al., 2025; Qian et al., 2024), while preserving overall linguistic fluency and reasoning logic, as illustrated in Figure 3. To ensure that the perturbations are faithfully reflected, we automatically verify each generated response by comparing its answer consistency with the ground-truth reference for the corresponding visual question, discarding cases where the perturbation fails to alter the intended attribute. This approach is motivated by the empirical observation that mul-

timodal LLMs exhibit higher controllability and precision when instructed to *generate* subtle perceptual inconsistencies rather than when tasked to *detect* them, enabling stable synthesis of realistic yet controlled visual errors (Jiang et al., 2024; Qian et al., 2024).

To further decouple perceptual and reasoning dimensions, we synthesize an additional response $r_{r_{p+r}}$ by introducing both *perceptual and reasoning inconsistencies* (p+r), resulting in a fully degraded sample. Each instance thus forms a quadruplet $(x_i, r_c, r_{r_p}, r_{r_{p+r}})$ representing (i) the given textual and visual input, (ii) the correct (chosen), (iii) the perceptually perturbed (rejected$_1$), and (iv) the perceptually and logically inconsistent responses (rejected$_2$).

The PPJD is designed to provide fine-grained supervision for verifiable judgment learning, enabling the judge to learn explicit ordering relationships such as $r_c \succ r_{r_p} \succ r_{r_{p+r}}$. In practice, this construction effectively achieves the intended guidance, and the resulting quadruplets serve as the foundation for the reward modeling and optimization framework described in Section 3.4. Further details of the data generation process and prompt design are provided in the Supplementary Material.

## 3.4. Verifiable Reward Modeling

We design a verifiable reward scheme (Wen et al., 2025; Su et al., 2025) that encourages perceptually grounded and human-aligned judgments while penalizing over-reliance on response-described cues or surface-level reasoning. The judge MLLM, $\mathcal{M}_{\text{Judge}}$, is trained with Group Relative Pol-

icy Optimization (GRPO) (Shao et al., 2024; DeepSeek-AI et al., 2025) using quadruplets $(x_i, r_c, r_{r_p}, r_{r_{p+r}})$, where $r_c$ is the verified chosen response, $r_{r_p}$ is the perception-perturbed rejected$_1$ response, and $r_{r_{p+r}}$ contains both perceptual and reasoning perturbations (rejected$_2$). Let $o_i$ denote the model-generated verdict (reasoning–answer output) for input $x_i$. GRPO optimizes relative advantages within grouped responses without requiring a value function, leading to stable and scalable alignment. This property aligns naturally with our setting, where each quadruplet forms a verifiable preference group exhibiting graded perceptual and reasoning degradation.

**Structural Reward.** Before evaluating perceptual or reasoning quality, the judge must produce verifiable outputs that follow a consistent reasoning–answer pattern. This step guarantees syntactic and semantic validity, ensuring that only well-formed responses contribute to subsequent rewards. The judge produces a textual verdict, $v$, that follows the required `<think>...</think>` and `<answer>...</answer>` structure while expressing either a pairwise preference or a three-way ranking. Let $y_i$ denote the model output for instance $i$ and define the format validator:

$$\mathcal{R}_{\text{Format}}(o_i) = \begin{cases} 1, & \text{if } y_i \text{ satisfies the required structure} \\ & \text{and value ranges,} \\ 0, & \text{otherwise.} \end{cases}$$

(5)

All content rewards are gated by $\mathcal{R}_{\text{Format}}$. If the format is invalid, the total reward for that instance is zero.

**Batch Ranking Reward.** While pairwise supervision captures local preference correctness, it cannot guarantee globally consistent ordering among all responses. We therefore incorporate a listwise ranking term that evaluates the entire triplet jointly, offering a denser and more informative supervision signal that encourages transitive consistency in the model's judgment behavior. We evaluate the predicted permutation $\hat{\boldsymbol{\pi}}_i$ over $\{r_c, r_{r_p}, r_{r_{p+r}}\}$ against the target $\boldsymbol{\pi}_i^{\star} = (r_c, r_{r_p}, r_{r_{p+r}})$ inspired by the weighted Levenshtein distance (Yujian & Bo, 2007) $d_{\text{Lev}}(\hat{\boldsymbol{\pi}}_i, \boldsymbol{\pi}_i^{\star})$ normalized by the maximum distance as:

$$\mathcal{R}_{\text{Batch}}(o_i) = 1 - d_{\text{Lev}}(\hat{\boldsymbol{\pi}}_i, \boldsymbol{\pi}_i^{\star}) / ||\boldsymbol{\pi}_i^{\star}||.$$

(6)

This scheme yields the following discrete rewards:

$$\mathcal{R}_{\text{Batch}}(o_i) = \begin{cases} 1, & r_c \succ r_{r_p} \succ r_{r_{p+r}}, \\ \frac{2}{3}, & r_c \succ r_{r_{p+r}} \succ r_{r_p} \text{ or } r_{r_p} \succ r_c \succ r_{r_{p+r}}, \\ \frac{1}{3}, & r_{r_p} \succ r_{r_{p+r}} \succ r_c \text{ or } r_{r_{p+r}} \succ r_c \succ r_{r_p}, \\ 0, & r_{r_{p+r}} \succ r_{r_p} \succ r_c. \end{cases}$$

(7)

To ensure uniform and interpretable scaling, the reward values are distributed evenly across discrete ranking correctness levels, assigning higher rewards to permutations closer to the target order. This formulation preserves consistent reward intervals without relying on task-specific heuristics.

**Unified Verifiable Reward and Optimization.** Finally, we integrate all components into a unified verifiable objective that combines structural validity and global ranking consistency. This hierarchical formulation ensures that the judge model produces structurally valid, perceptually faithful, and human-aligned judgments across perturbation scales. The total per-instance reward is a format-gated product of the batch component as:

$$\mathcal{R}(o_i) = \mathcal{R}_{\text{Format}}(o_i) \times \mathcal{R}_{\text{Batch}}(o_i),$$

(8)

where $\mathcal{R}_{\text{Format}}(o_i) \in \{0, 1\}$ gates invalid outputs.

We train the judge model using Group Relative Policy Optimization (GRPO) (Shao et al., 2024; DeepSeek-AI et al., 2025), where the unified verifiable reward $\mathcal{R}(o_i)$ defined in Equation (8) is incorporated into the GRPO objective in Equation (1). This joint formulation allows the model to align its judgment behavior with perceptually grounded supervision in a stable and scalable manner.

## 4. Experiments

### 4.1. Experimental Setup

**Datasets.** We construct the PPJD training set using MMPR v1.2 (Wang et al., 2024). This dataset provides multimodal chosen and rejected preference pairs spanning eleven reasoning categories, including general VQA, science, mathematics, chart understanding, and document reasoning. To avoid data leakage, we remove all samples that overlap with the evaluation split. Following the procedure described in Section 3.3, we extract 3k high-quality pairs to form the PPJD training set. Additional statistics and dataset construction details are provided in the supplementary material (Section D). For evaluation, consistent with prior work (Chen et al., 2024; Xiong et al., 2025; Ko et al., 2025), we assess the judge model using the MLLM-as-a-Judge benchmark (Chen et al., 2024), which comprises 14 diverse vision-language tasks covering general VQA, mathematical reasoning, and scientific knowledge, enabling a comprehensive assessment of both generality and robustness.

**Implementation Details.** We implement Perception-Judge-Flex and Perception-Judge-Qwen3 based on the Flex-Judge-VL-7B (Ko et al., 2025) and Qwen3-VL-4B-Thinking (Yang et al., 2025) architectures, respectively. All models are trained using the `verl` framework (Sheng et al., 2024). Additional training configurations and implementation details are provided in the supplementary material (Section C).

**Evaluation.** To evaluate the performance of judge models, we measure how closely the judge model's assessments align with human annotations. For single- score

*Table 2.* **The overall comparison of MLLM judging performance against human annotations across different datasets.** Each judgment is sampled three times and averaged to reduce variance. "w." and "w.o." indicate evaluation with and without ties. †: reported results from LLaVA-Critic (Xiong et al., 2025). *: results from MLLM-as-a-Judge (Chen et al., 2024). Upon consultation with the authors, we utilized our reproduced numbers for Flex-Judge-VL (Ko et al., 2025) to ensure a consistent and fair comparison.

| | Model | Size | COCO | C.C. | Diff. | Graphics | Math | Text | WIT | Chart | VisIT | CC-3M | M2W | SciQA | Aes | MM-Vet | Avg. |
|---|---|---|---|---|---|---|---|---|---|---|---|---|---|---|---|---|---|
| Score (↑) | GPT-4V † | - | 0.410 | 0.444 | 0.361 | 0.449 | 0.486 | 0.506 | 0.457 | 0.585 | 0.554 | 0.266 | 0.267 | 0.315 | 0.472 | 0.367 | 0.424 |
| | GPT-4o † | - | 0.396 | 0.452 | 0.341 | 0.464 | 0.460 | 0.564 | 0.408 | 0.573 | 0.589 | 0.305 | 0.262 | 0.569 | 0.421 | 0.342 | 0.439 |
| | Gemini-1.0-Pro-Vision * | - | 0.262 | 0.408 | - | 0.400 | 0.228 | 0.222 | 0.418 | 0.343 | 0.336 | 0.374 | 0.324 | 0.073 | 0.360 | 0.207 | 0.304 |
| | Prometheus-V † | 13B | 0.289 | 0.342 | 0.106 | 0.172 | 0.182 | 0.214 | 0.209 | 0.224 | 0.226 | 0.228 | 0.089 | 0.174 | 0.368 | 0.157 | 0.213 |
| | LLaVA-1.5 * | 13B | 0.247 | 0.227 | 0.060 | 0.242 | 0.093 | 0.245 | 0.109 | 0.237 | 0.177 | 0.071 | 0.424 | 0.279 | 0.414 | 0.322 | 0.225 |
| | LLaVA-1.6 * | 34B | 0.285 | 0.251 | -0.012 | 0.262 | 0.238 | 0.258 | 0.151 | 0.318 | 0.198 | 0.109 | 0.022 | 0.206 | 0.025 | 0.265 | 0.184 |
| | LLaVA-Critic † | 7B | 0.382 | 0.450 | 0.103 | 0.316 | 0.356 | 0.378 | 0.179 | 0.421 | 0.322 | 0.246 | 0.301 | 0.269 | 0.395 | 0.272 | 0.314 |
| | Qwen2.5-VL-Instruct | 7B | 0.294 | 0.247 | -0.020 | -0.041 | 0.095 | 0.170 | 0.056 | 0.011 | 0.328 | 0.178 | 0.255 | 0.311 | 0.327 | 0.103 | 0.165 |
| | Flex-Judge-VL | 7B | 0.201 | 0.297 | 0.049 | 0.301 | 0.468 | 0.437 | 0.111 | 0.513 | 0.381 | 0.189 | 0.321 | 0.361 | 0.296 | 0.311 | 0.404 |
| | Qwen3-VL-Thinking | 4B | 0.342 | 0.392 | 0.271 | 0.405 | 0.353 | 0.489 | 0.284 | 0.607 | 0.388 | 0.225 | 0.074 | 0.521 | 0.319 | 0.308 | 0.419 |
| | Perception-Judge-Flex | 7B | 0.293 | 0.277 | 0.140 | 0.358 | 0.487 | 0.429 | 0.164 | 0.545 | 0.447 | 0.225 | 0.366 | 0.399 | 0.401 | 0.374 | 0.466 |
| | Perception-Judge-Qwen3 | 4B | 0.348 | 0.410 | 0.347 | 0.401 | 0.401 | 0.489 | 0.225 | 0.609 | 0.462 | 0.273 | 0.142 | 0.556 | 0.344 | 0.332 | 0.457 |
| Pair w. Tie (↑) | GPT-4V † | - | 0.539 | 0.634 | 0.668 | 0.632 | 0.459 | 0.495 | 0.536 | 0.369 | 0.591 | 0.544 | 0.544 | 0.389 | 0.620 | 0.517 | 0.538 |
| | GPT-4o † | - | 0.539 | 0.634 | 0.668 | 0.632 | 0.459 | 0.495 | 0.536 | 0.369 | 0.591 | 0.544 | 0.544 | 0.389 | 0.620 | 0.517 | 0.538 |
| | Gemini-1.0-Pro-Vision * | - | 0.616 | 0.787 | - | 0.650 | 0.436 | 0.664 | 0.605 | 0.500 | 0.660 | 0.560 | 0.370 | 0.262 | 0.190 | 0.312 | 0.509 |
| | LLaVA-1.5 * | 13B | 0.273 | 0.478 | 0.286 | 0.273 | 0.657 | 0.510 | 0.369 | 0.383 | 0.456 | 0.484 | 0.347 | 0.223 | 0.389 | 0.254 | 0.384 |
| | LLaVA-Critic † | 7B | 0.593 | 0.687 | 0.707 | 0.587 | 0.432 | 0.544 | 0.564 | 0.338 | 0.596 | 0.628 | 0.591 | 0.370 | 0.686 | 0.464 | 0.556 |
| | Qwen2.5-VL-Instruct | 7B | 0.446 | 0.474 | 0.507 | 0.326 | 0.397 | 0.383 | 0.366 | 0.364 | 0.461 | 0.483 | 0.358 | 0.442 | 0.494 | 0.420 | 0.423 |
| | Flex-Judge-VL | 7B | 0.504 | 0.624 | 0.611 | 0.552 | 0.454 | 0.518 | 0.413 | 0.403 | 0.565 | 0.562 | 0.588 | 0.389 | 0.589 | 0.507 | 0.514 |
| | Qwen3-VL-Thinking | 4B | 0.547 | 0.653 | 0.743 | 0.559 | 0.48 | 0.495 | 0.613 | 0.367 | 0.586 | 0.595 | 0.593 | 0.347 | 0.666 | 0.493 | 0.543 |
| | Perception-Judge-Flex | 7B | 0.541 | 0.671 | 0.567 | 0.562 | 0.454 | 0.511 | 0.444 | 0.384 | 0.577 | 0.592 | 0.554 | 0.367 | 0.629 | 0.482 | 0.520 |
| | Perception-Judge-Qwen3 | 4B | 0.541 | 0.661 | 0.777 | 0.593 | 0.489 | 0.510 | 0.605 | 0.369 | 0.583 | 0.604 | 0.613 | 0.352 | 0.695 | 0.504 | 0.554 |
| Pair w.o. Tie (↑) | GPT-4V † | - | 0.729 | 0.772 | 0.884 | 0.853 | 0.665 | 0.661 | 0.760 | 0.495 | 0.785 | 0.707 | 0.697 | 0.639 | 0.741 | 0.654 | 0.717 |
| | GPT-4o † | - | 0.774 | 0.776 | 0.934 | 0.835 | 0.628 | 0.618 | 0.737 | 0.513 | 0.741 | 0.770 | 0.706 | 0.722 | 0.887 | 0.660 | 0.736 |
| | Gemini-1.0-Pro-Vision * | - | 0.717 | 0.840 | - | 0.770 | 0.678 | 0.793 | 0.688 | 0.658 | 0.711 | 0.652 | 0.471 | 0.358 | 0.265 | 0.400 | 0.615 |
| | LLaVA-1.5 * | 13B | 0.327 | 0.537 | 0.302 | 0.300 | 0.726 | 0.684 | 0.600 | 0.610 | 0.648 | 0.583 | 0.449 | 0.443 | 0.498 | 0.344 | 0.504 |
| | LLaVA-Critic † | 7B | 0.771 | 0.774 | 0.755 | 0.758 | 0.596 | 0.658 | 0.680 | 0.488 | 0.727 | 0.742 | 0.692 | 0.658 | 0.715 | 0.635 | 0.689 |
| | Qwen2.5-VL-Instruct | 7B | 0.479 | 0.492 | 0.510 | 0.268 | 0.368 | 0.394 | 0.334 | 0.348 | 0.506 | 0.538 | 0.330 | 0.511 | 0.486 | 0.388 | 0.425 |
| | Flex-Judge-VL | 7B | 0.644 | 0.691 | 0.634 | 0.693 | 0.573 | 0.595 | 0.491 | 0.564 | 0.667 | 0.656 | 0.651 | 0.660 | 0.600 | 0.630 | 0.623 |
| | Qwen3-VL-Thinking | 4B | 0.708 | 0.726 | 0.776 | 0.696 | 0.604 | 0.567 | 0.724 | 0.513 | 0.699 | 0.697 | 0.660 | 0.606 | 0.682 | 0.656 | 0.663 |
| | Perception-Judge-Flex | 7B | 0.703 | 0.746 | 0.592 | 0.715 | 0.604 | 0.594 | 0.532 | 0.546 | 0.697 | 0.694 | 0.622 | 0.665 | 0.646 | 0.652 | 0.645 |
| | Perception-Judge-Qwen3 | 4B | 0.707 | 0.737 | 0.815 | 0.749 | 0.649 | 0.597 | 0.726 | 0.532 | 0.706 | 0.708 | 0.689 | 0.663 | 0.716 | 0.717 | 0.691 |
| Batch (↓) | GPT-4V † | - | 0.318 | 0.353 | 0.070 | 0.385 | 0.348 | 0.319 | 0.290 | 0.347 | 0.300 | 0.402 | 0.597 | 0.462 | 0.453 | 0.411 | 0.361 |
| | Gemini-1.0-Pro-Vision * | - | 0.287 | 0.299 | - | 0.473 | 0.462 | 0.430 | 0.344 | 0.520 | 0.426 | 0.357 | 0.613 | 0.412 | 0.467 | 0.529 | 0.432 |
| | LLaVA-1.5 * | 13B | 0.577 | 0.492 | 0.562 | 0.535 | 0.598 | 0.650 | 0.616 | 0.644 | 0.620 | 0.563 | 0.639 | 0.563 | 0.650 | 0.652 | 0.597 |
| | Qwen2.5-VL-Instruct | 7B | 0.562 | 0.450 | 0.593 | 0.630 | 0.607 | 0.582 | 0.631 | 0.570 | 0.569 | 0.519 | 0.639 | 0.703 | 0.558 | 0.572 | 0.585 |
| | Flex-Judge-VL | 7B | 0.487 | 0.462 | 0.531 | 0.559 | 0.505 | 0.543 | 0.576 | 0.482 | 0.486 | 0.537 | 0.526 | 0.551 | 0.521 | 0.478 | 0.517 |
| | Qwen3-VL-Thinking | 4B | 0.532 | 0.469 | 0.401 | 0.535 | 0.514 | 0.513 | 0.507 | 0.452 | 0.544 | 0.539 | 0.534 | 0.453 | 0.488 | 0.479 | 0.498 |
| | Perception-Judge-Flex | 7B | 0.467 | 0.428 | 0.487 | 0.535 | 0.492 | 0.516 | 0.547 | 0.529 | 0.456 | 0.528 | 0.540 | 0.560 | 0.541 | 0.435 | 0.505 |
| | Perception-Judge-Qwen3 | 4B | 0.447 | 0.371 | 0.296 | 0.491 | 0.448 | 0.436 | 0.415 | 0.426 | 0.432 | 0.521 | 0.578 | 0.431 | 0.479 | 0.434 | 0.444 |

grading tasks, we quantify this alignment using the Pearson correlation (Lee Rodgers & Nicewander, 1988). For pairwise comparisons, we measure the correspondence between model and human decisions using accuracy, F1 score, and recall (Goutte & Gaussier, 2005). For batch -level evaluation, we consolidate the ranking outputs into a single sequence and compute their similarity to human judgments using the normalized Levenshtein distance (Levenshtein, 1966).

**Baselines.** We compare our Perception-Judge against competitive commercial proprietary MLLMs with high API usage costs (Achiam et al., 2023; Comanici et al., 2025), as well as open-source models that either require substantial curated training sets (Lee et al., 2024; Xiong et al., 2025) or rely on significantly larger model sizes (Liu et al., 2023).

## 4.2. Performance Evaluation

**Quantitative Results.** We evaluate our fine-tuned multimodal judge on the MLLM-as-a-Judge benchmark as summarized in Table 2. Our model shows consistent improvements across most evaluation settings. Specifically, our method exhibits substantial gains over the state-of-the-art Qwen3-VL-4B-Thinking (Yang et al., 2025), showing improvements of 4% in pairwise evaluations and 11% in batch-level evaluations. Moreover, even though no explicit score labels are used during training, our method yields a remarkable increase of 12% in single-score grading tasks. This suggests that our model effectively learns the scalar evaluation capability through relative preference reward modeling.

Overall, our fine-tuned model not only exceeds existing

open-source MLLMs but also achieves performance comparable to leading proprietary LLM-based evaluators such as GPT-4o (Hurst et al., 2024) in single-score prediction mode. Notably, our model also outperforms most proprietary LLMs in the batch-level evaluation, where the model demonstrates stronger global consistency across multi-response comparisons.

We attribute these improvements to our relative concept-aware training dataset, which encourages the model to reason over inter-sample relations rather than relying on superficial patterns. To validate the reduction in bias, we analyze the model's accuracy in identifying corrected responses against perception-perturbed ones in Figure 2. The results quantitatively demonstrate that our approach effectively mitigates perceptual and reasoning biases.

This enhanced robustness is further corroborated by our results on InfographicVQA and MathVista, where our model achieves consistent gains across all three evaluation settings. Given that these benchmarks necessitate precise optical character recognition (OCR) and mathematical reasoning, these improvements strongly suggest that our training strategy successfully enhances the model's capacity for perception-related reasoning.

While LLaVA-Critic (Xiong et al., 2025) attains higher scores in the pairwise setting, it is trained on the LLaVA-Critic-113k corpus. In contrast, our model is fine-tuned on a compact 3k dataset that is carefully curated to be fully disjoint from all evaluation benchmarks. This demonstrates the exceptional data efficiency of our approach, achieving competitive performance with significantly less supervision.

We provide extended evaluations on a broader range of models in the supplementary material (Section F.1), demonstrating the generalizability of our approach.

**Qualitative Results.** The improvements in the perceptual judgment, reasoning behavior, and evaluation consistency of our method are shown in Figure 4. We provide additional examples in the supplementary material (Section G).

### 4.3. Ablation Study

In this section, we address two questions: (1) What advantages does our proposed PPJD dataset (Section 3.3) offer over the existing MMPR-v1.2 (Wang et al., 2024) dataset during training? (2) How do the reward designs introduced in Section 3.4 affect model performance? To this end, we fine-tune Flex-Judge-VL-7B under three configurations, each controlled to utilize 10k training samples for fair comparison: (i) MMPR-v1.2 (Wang et al., 2024) with pairwise reward, (ii) PPJD with pairwise reward, and (iii) PPJD with batch ranking reward. For pairwise reward, we use a binary reward that assigns 1 when the chosen response receives a higher score than its counterpart, and 0 otherwise.

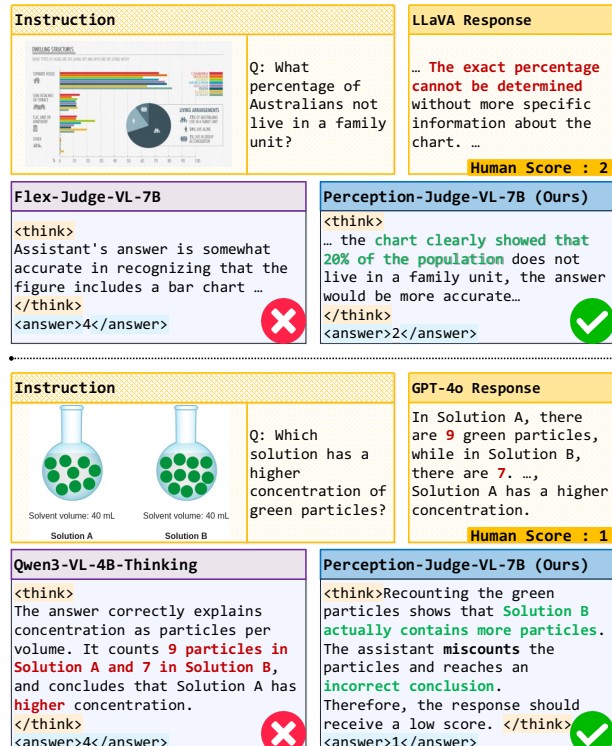

*Figure 4.* **A qualitative comparison between our Perception-Judge and the baseline (Flex-Judge and Qwen3-VL-4B-Thinking).** While the baseline gives a high score to a response that fails to incorporate the visual content, our model interprets the relevant visual cue into its reasoning and penalizes their absence in the final score.

For PPJD, we construct preference pairs as $(r_c, r_{r_p})$ and $(r_{r_p}, r_{r_{p+r}})$, and apply the same pairwise reward, enabling the model to learn from both corrected–perceptual and perceptual–perturbed comparisons.

*Table 3.* **Ablation study on the choice of fine-tuning dataset and reward type.** The Overall performance for four configurations: (1) Base model, (2) MMPR-v1.2 with pairwise reward, (3) PPJD with pairwise reward, and (4) PPJD with batch ranking reward.

| Dataset | Type | Score (↑) | Pair w. Tie (↑) | Pair w.o. Tie (↑) | Batch (↓) |
|---------|------|-----------|-----------------|-------------------|-----------|
| – | – | 0.404 | 0.514 | 0.623 | 0.517 |
| MMPR-v1.2 | Pair | 0.454 | 0.515 | 0.641 | 0.515 |
| PPJD | Pair | 0.458 | **0.518** | 0.644 | 0.513 |
| PPJD | Batch | **0.476** | **0.518** | **0.648** | **0.500** |

**Effect of Datasets.** To isolate the contribution of PPJD, we compare MMPR-v1.2 with pairwise reward (2nd row) against PPJD with pairwise reward (3rd row). Replacing MMPR-v1.2 with PPJD for training data yields consistent improvements across all metrics. Since PPJD explicitly encodes perceptual corrections through $(r_c, r_{r_p}, r_{r_{p+r}})$, these gains indicate that directly mitigating perceptual judgment bias in the training data leads to more reliable multimodal evaluation than training solely on generic (chosen, rejected) pairs.

**Effect of Rewards.** To compare the effect of rewards, we examine PPJD with pairwise reward (3rd row) against PPJD with batch ranking reward (4th row). Using only the batch ranking objective yields higher performance on both score-based and pairwise metrics. This is particularly notable because the batch reward never observes explicit pairwise or score labels; instead, it enforces a globally consistent ordering of candidates within each batch. These results suggest that global ranking supervision is sufficient not only for improving set-level rankings, but also for inducing accurate local preferences and well-calibrated score predictions.

## 5. Conclusion

This work systematically investigates perceptual reliability in multimodal LLM-based judges and identifies **Perceptual Judgment Bias**, a fundamental failure mode in which models prioritize linguistic plausibility over visual correctness, leading to systematic divergence from human evaluation. To address this limitation, we propose a perception-guided training framework built upon the **Perceptually Perturbed Judgment Dataset (PPJD)** and a unified GRPO-based optimization with **structured batch reward modeling**, enabling fine-grained, verifiable supervision and globally consistent ranking without explicit pairwise labels. Extensive experiments across diverse benchmarks demonstrate substantial improvements in perceptual fidelity, ranking coherence, and alignment with human judgments, establishing a principled foundation for robust, interpretable, and scalable multimodal evaluation.

## Impact Statement

This work aims to advance the field of machine learning by improving the reliability and fairness of automated multimodal evaluators. By mitigating perceptual judgment bias, our framework supports the development of evaluation systems that are more transparent, better grounded in visual evidence, and more closely aligned with human judgment. This advancement is increasingly relevant as MLLMs are applied in settings where accurate visual understanding and faithful evaluation are critical.

More broadly, potential societal risks arise if automated judges replace human oversight in consequential decision-making. Although our approach reduces the cost of evaluation and the reliance on extensive human annotation, residual biases and failure modes may persist, particularly in subjective, ambiguous, or out-of-distribution cases. These risks could be amplified if such systems are treated as definitive authorities. We therefore recommend deploying MLLM-as-a-Judge systems as decision-support tools, complemented by human review and domain-appropriate auditing in high-stakes environments.

## Acknowledgments

This work was supported by Samsung Research, Samsung Electronics Co., Ltd.; the Basic Science Research Program through the National Research Foundation of Korea (NRF) funded by the Korea government (MSIT) (No. RS-2025-00520207); Institute of Information & Communications Technology Planning & Evaluation (IITP) grants funded by the Korea government (MSIT) (Nos. RS-2024-00457882, 2022-0-01045, 2022-0-00680, RS-2019-II190075); the Advanced GPU Utilization Support Program funded by the Government of the Republic of Korea (Ministry of Science and ICT); the AI Computing Infrastructure Enhancement (GPU Rental Support) User Support Program funded by the Ministry of Science and ICT (MSIT), Republic of Korea (No. RQT-25-120217); and a grant partly supported by both IITP (MSIT) and Korea Evaluation Institute of Industrial Technology (KEIT) (MOTIE) (No. RS-2025-02217259).

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

# A. Limitations & Future Work

Although our approach demonstrates competitive performance and presents clear potential for practical multimodal evaluation, several aspects remain open for further exploration. First, the current judge performance still trails behind expert human annotators in certain challenging scenarios. This gap suggests an opportunity to refine the reasoning procedure and expand the diversity of preference signals used during training. Second, the training method requires larger computation compared with approaches such as DPO, especially when operating over multiple sampling rounds. Reducing computational overhead while maintaining robustness would further increase the practicality of our framework.

Another limitation is the dependency on proprietary models for generating supervisory signals. While this choice enables high-quality guidance, future work could explore self-distillation (Yuan et al., 2024; Sun et al., 2023; Saha et al., 2025) using capable open-source models, such as Qwen (Bai et al., 2023). This direction has the potential to improve reproducibility and broaden applicability in settings where access to closed models is restricted. Extending the method to larger model scales also remains an important step.

In addition, certain aspects of subjective evaluation remain challenging. Although the proposed procedure reduces variability, subtle ambiguities in multimodal interpretation can still occur, and residual noise in the reasoning process may influence outcomes even after filtering.

Overall, our method lays a strong foundation for multimodal evaluation and opens several promising paths for future development.

# B. Related Work

**LLM-as-a-Judge.** The *LLM-as-a-Judge* paradigm (Chiang et al., 2024; Li et al., 2024, 2025a; Thakur et al., 2025; Whitehouse et al., 2025; Saha et al., 2025; Chen et al., 2025a; Zhu et al., 2023) employs language models as evaluators of responses produced by other LLMs. Models such as Prometheus (Kim et al., 2024; Lee et al., 2024) provide direct scoring or pairwise comparison. Recent works (Whitehouse et al., 2025; Chen et al., 2025a) further incorporate reinforcement learning (RL) with verifiable reward schemes (Su et al., 2025; Wen et al., 2025; Yuan et al., 2024), typically composed of (1) the difference between model-assigned and human-annotated scores (Chen et al., 2025a), (2) correctness and consistency of pairwise preferences (Whitehouse et al., 2025; Saha et al., 2025), and (3) structural rewards ensuring valid output formats or score ranges.

The *MLLM-as-a-Judge* framework (Chen et al., 2024; Ko et al., 2025; Pu et al., 2025; Waheed et al., 2025; Lee et al., 2024; Xiong et al., 2025) extends this idea to multimodal evaluation, enabling models to assess vision–language responses. However, existing approaches mainly rely on supervised fine-tuning rather than RL-based optimization, and current multimodal judges often exhibit *visual perception judgment bias*, favoring linguistic plausibility over perceptual correctness. Our work addresses this limitation by introducing an RL-based training framework that explicitly reinforces perception-grounded judgment through verifiable rewards.

**Visual Perception and Visual Reasoning.** Recent studies have revealed that MLLMs often struggle with visual reasoning tasks due to their limited visual perception capabilities (Chen et al., 2025b; Jian et al., 2025; Tian et al., 2025; Wang et al., 2025b; Chen et al., 2025c; Bigverdi et al., 2025). A series of works identifies this limitation as stemming from the under-attention to visual information when the textual sequence length dominates the number of visual tokens. ADAPTVIS (Chen et al., 2025b) addresses this issue by dynamically re-weighting the attention of visual tokens based on confidence criteria, improving visual perception in a training-free manner. Reflection-V (Jian et al., 2025) adopts an RL approach that uses rewards derived from visual-token attention values, promoting reasoning grounded in visual information. VAPO (Tian et al., 2025) augments reasoning datasets by randomly inserting visual perception questions into textual reasoning sequences and trains the model with RL to answer them, encouraging greater attention to visual information during reasoning.

Another line of research develops complementary approaches to enhance visual perception. PAPO (Wang et al., 2025b) collects negative responses generated from corrupted image inputs to help models distinguish correct reasoning from perceptually degraded reasoning. MiCo (Chen et al., 2025c) teaches MLLMs to discriminate whether an image is an augmented version of the original, thereby improving visual reasoning ability. LLaVA-AURORA (Bigverdi et al., 2025) proposes perception tokens, which embed intrinsic image features as reasoning tokens to enhance visual reasoning. Unlike prior studies that primarily enhance perceptual grounding within the reasoning process, our work focuses on elevating perception itself in the context of multimodal judgment.

# C. Experimental Details

We provide additional experimental information to facilitate reproducibility and clarify the computational configuration used in this work. All experiments, including GRPO-based fine-tuning and evaluation across multimodal judgment benchmarks, were conducted using the same standard-

ized setup unless otherwise specified.

### C.1. Code & Reproduction

The full training and evaluation pipeline is implemented in PyTorch and built upon the `verl` framework (Sheng et al., 2024) for reinforcement learning with verifiable rewards. The code is available at the GitHub repository (https://github.com/kaist-cvml/perception-judge).

### C.2. Implementation Details & Hyperparameters

Perception-Judge is initialized from two models: Flex-Judge-VL and Qwen3-VL-Thinking. Both models are optimized using Group Relative Policy Optimization (GRPO) with five rollouts per prompt. Training is conducted for one epoch on the PPJD dataset with a global batch size of 512. We employ a fixed learning rate of $2.5 \times 10^{-7}$ for Flex-Judge-VL and $7.5 \times 10^{-7}$ for Qwen3-VL-Thinking. All reward components and structural validators follow the formulations described in the main paper. The maximum sequence length is set to 4,096 tokens for prompts and 24,576 tokens for responses. For GRPO, we use clipping with $\epsilon = 0.2$ and set the KL coefficient to $0.01$. Throughout all experiments, we use a temperature of $1.0$ and top-p of $0.95$ for sampling.

We employed 8 NVIDIA H100 GPUs for training Perception-Judge-Flex and 8 NVIDIA H200 GPUs for Perception-Judge-Qwen3. Inference follows the same model configuration without additional tuning and uses the standardized evaluation protocol defined in the MLLM-as-a-Judge benchmark (Chen et al., 2024). Additional details on the construction of the dataset are provided in Section D.

### C.3. Evaluation Details

#### C.3.1. EVALUATION SETTINGS

In this section, we describe the evaluation protocol used in our experiments, which follows the practices commonly adopted in MLLM-as-a-Judge research (Chen et al., 2024; Xiong et al., 2025; Ko et al., 2025). Our goal is to assess how closely a judge model aligns with human preferences (Judgement GT) across three fundamental evaluation settings: single-score grading, pairwise comparison, and batch-level ranking. The protocol integrates established practices from prior work, including MLLM-as-a-Judge (Chen et al., 2024), Flex-Judge (Ko et al., 2025), and related multimodal and LLM evaluation benchmarks (Pu et al., 2025; Li et al., 2025a).

**Single-Score Grading.** Given an input $x = (q, v)$ consisting of a textual query $q$ and its associated multimodal context $v$, the judge model assigns a scalar score $s \in [1, 5]$ for a candidate response $r$. Following prior benchmarks, agreement with human ratings is measured using Pearson

---

```
You are a helpful assistant. The assistant first
performs a detailed, step-by-step reasoning
process in its mind and then provides the user
with the answer.

Now the user asks you to judge the performance of
an AI assistants in response to the question.

Score assistants 1-10 (higher=better). Criteria
includes helpfulness, relevance, accuracy, and
level of detail. Avoid length, style, or other
bias.
```

*Figure S1.* The shared prompt template used across the three evaluation settings (single-score, pairwise comparison, and batch ranking).

correlation. Formally,

$$s = \mathcal{M}_{\text{Judge}}(x, r). \tag{S1}$$

**Pairwise Comparison.** Given two candidate responses $r_a$ and $r_b$, the judge outputs a preference relation $r_a \succ r_b$, $r_b \succ r_a$, or (optionally) a tie. Accuracy is measured by comparing the predicted preference to human-annotated labels. To mitigate position bias, the response ordering is randomized during evaluation as in (Chen et al., 2024; Ko et al., 2025).

**Batch-Level Ranking.** For a set of $K$ model outputs $\{r_k\}_{k=1}^{K}$, the judge produces a ranking sequence based on descending judgment scores. The predicted permutation is compared to the ground-truth human ranking using the normalized Levenshtein distance (Yujian & Bo, 2007), following standard practice (Chen et al., 2024). Let $\pi$ denote the predicted permutation, and $\pi^*$ the human annotation. The normalized distance (or inverse similarity) is computed as:

$$D_{\text{batch}} = \frac{d_{\text{Lev}}(\pi, \pi^*)}{K}. \tag{S2}$$

#### C.3.2. EVALUATION PROMPTS

For completeness, we include the prompt formats used across the three evaluation settings. We first present the common prompt Figure S1 template shared by all settings, followed by the specific variants used for each evaluation protocol. The actual input content is replaced with placeholders as requested.

**Single-Score Prompt.** The prompt template used for single-score grading is shown in Figure S2.

---

**Evaluation: SINGLE-SCORE PROMPT**

```
After thinking, when you finally reach a
conclusion, clearly provide your evaluation
scores within <answer></answer> tags,
i.e., for example, <answer>3</answer>
```

*Figure S2.* Prompt template used for the single-score evaluation.

---

**Evaluation: PAIRWISE COMPARISON PROMPT**

```
After thinking, when you finally reach a
conclusion, clearly provide your evaluation
scores within <answer></answer> tags, i.e.,
for example,<answer>3</answer><answer>5</answer>
```

*Figure S3.* Prompt template used for the pairwise comparison.

---

**Evaluation: BATCH RANKING PROMPT**

```
DO NOT assign the same score to multiple
assistants. After thinking, when you finally
reach a conclusion, clearly provide your
evaluation scores within <answer> </answer> tags,
i.e., for example, <answer>3</answer><answer>8
</answer><answer>6</answer>
```

*Figure S4.* Prompt template used for the batch ranking evaluation.

The prompt template used for single-score grading is shown in Figure S2. Although the MLLM outputs scores on a 1 to 10 scale, we apply a simple deterministic post-processing step (simple division) to map them to the 1 to 5 range used in our evaluation.

**Pairwise Comparison Prompt.** The prompt format used for pairwise evaluations is provided in Figure S3. After the answer generation, we apply a simple post-processing step that determines the preference relation by comparing the two scores or assigning a tie when they fall within the same range.

**Batch Ranking Prompt.** The prompt design used for batch-level ranking is illustrated in Figure S4. A lightweight post-processing step then converts the model's assigned scores into an ordered ranking over the candidate set.

### C.3.3. BIAS MITIGATION

Consistent with prior evaluations (Chen et al., 2024; Ko et al., 2025), we apply two strategies to reduce judgment bias:

- **Order Randomization** to reduce positional bias in pairwise and batch settings.

- **Tie Handling** when the benchmark includes a tie option, preventing forced-choice artifacts; when a tie option is not available, the model is required to make a

forced random choice between the candidates.

### C.4. Baselines

- **LLaVA-Critic.** LLaVA-Critic (Xiong et al., 2025) is an evaluation-centric 7B multimodal model, fine-tuned on a curated dataset of synthetic critiques. The model is designed to output scalar scores or pairwise preferences and demonstrates robust efficacy in factuality verification. However, it is important to note that potential overlaps between its training corpus and current evaluation benchmarks may lead to performance inflation. Despite this limitation, it remains a valuable open-source baseline for multimodal evaluation.

- **Qwen2.5-VL.** Qwen2.5-VL (Bai et al., 2023) is an open-source vision–language model family (0.5B–72B) with strong OCR and scene understanding. The 7B version integrates a ViT encoder with an LLaMA-based LLM, enabling robust VQA and multimodal reasoning. Although not trained as a judge, it performs reliable zero-shot evaluations through instruction prompting.

- **Flex-Judge-VL.** Flex-Judge-VL (Ko et al., 2025) is a 7B multimodal model trained exclusively on high-quality text-based reasoning demonstrations. Building upon the paradigm of text-centric evaluators like JudgeLM (Zhu et al., 2023), it generalizes to visual evaluation tasks despite the absence of image-specific supervision. By effectively leveraging multimodal representations to transfer reasoning capabilities, Flex-Judge-VL achieves performance competitive with large-scale proprietary models. This demonstrates the efficacy of transferring text-based reasoning patterns to the multimodal domain.

- **Prometheus-V.** Prometheus-V (Lee et al., 2024) is an open 13B evaluator trained on rubric-based multimodal feedback data. It provides interpretable scalar scores with detailed rationales. Although specialized for single-answer scoring rather than pairwise ranking, it offers consistent, fine-grained evaluation quality.

- **Qwen3-VL.** Qwen3-VL (Yang et al., 2025) is the latest open-source vision-language model family in the Qwen series, supporting unified interleaved inputs of text, images, and videos with a native long-context window up to 256K tokens. The model introduces architectural upgrades for stronger spatial-temporal grounding and vision-language alignment, leading to improved performance on multimodal reasoning benchmarks (e.g., multi-image and video understanding). While Qwen3-VL is not explicitly trained as an evaluator or judge, it can serve as a strong general-purpose multimodal backbone for zero-shot

or prompt-based assessment, benefiting from enhanced OCR, long-document/video comprehension, and robust instruction-following capabilities.

## D. Datasets Details

### D.1. PPJD Statistics

The Perceptually Perturbed Judgment Dataset (PPJD) consists of 3K instances and covers a set of multimodal reasoning categories that reflect the broader distribution of visual judgment tasks encountered in MLLM evaluation. The collected samples span general visual question answering, multi-step reasoning, mathematical and geometric queries, document-level information extraction, and image quality assessment. Each category contributes complementary visual signals and response patterns, enabling the dataset to capture both high-level semantic understanding and fine-grained perception-driven distinctions. This composition provides a representative basis for analyzing perceptual judgment behavior and serves as the foundation for the data generation procedure described in Section D.2. Detailed statistics and category-level counts are provided in Table S1.

*Table S1.* Category-level statistics of the Perceptually Perturbed Judgment Dataset (PPJD).

| Category | Included Datasets | Size | (%) |
|---|---|---|---|
| General VQA | VSR, TallyQA, OKVQA, GQA | 1.7 K | 45 % |
| Science | M³CoT | 0.6 K | 15 % |
| Mathematics | MAVIS, GeomVerse | 0.5 K | 14 % |
| OCR | SROIE | 0.5 K | 14 % |
| Chart | MapQA | 0.3 K | 7 % |
| Image Quality | KonIQ-10k | 0.2 K | 5 % |

### D.2. Detailed Dataset Generation Pipeline

We construct the Perceptually Perturbed Judgment Dataset (PPJD) dataset in a multi-stage pipeline starting from the MMPR corpus (Wang et al., 2024). Each example in PPJD is a quadruple $(x, r_c, r_{r_p}, r_{r_{p+r}})$, where $x$ denotes the visual (image) and instruction pair, $r_c$ is the original high quality answer from MMPR, and the two rejected candidates are generated by controlled perturbations of perceptual and reasoning attributes.

Figure 3 presents an overview of the PPJD construction pipeline. It summarizes the sequential stages used to transform raw MMPR samples into structured judgment quadruples, highlighting where perceptual and reasoning perturbations are introduced. In the following, we describe each stage in detail.

**Step 1. Source examples from MMPR and an absolute correctness check.** We begin by sampling image-instruction-answer (GT) triples from MMPR. Each MMPR

---

> Dataset Generation Step 1:
> VERIFY ANSWER PROMPT
> ```
> Check if CHOSEN matches ANSWER_GT given the image.
>
> QUESTION: { question }
> ANSWER_GT: { answer_gt }
> CHOSEN: { chosen }
>
> Output ONLY this JSON (no markdown, no extra
> text): {{ "is_correct": true }}
> or {{ "is_correct": false }}
> ```

*Figure S5.* The prompt template used to collect correct responses $r_c$ from the MMPR dataset. Starting from the preferred responses in MMPR, we filter the responses that match the ground-truth label, as verified by the MLLM (e.g., GPT-5). A response is accepted if the model returns {{"is_correct": true}}.

item provides a pairwise preference indicating which answer is relatively preferred, but this preferred response is not guaranteed to be absolutely correct. Therefore, before treating the preferred answer as the positive candidate $r_c$, we perform an additional correctness validation step.

Specifically, we compare the MMPR preferred answer against the ground-truth (GT) label associated with the original dataset from which the MMPR example was derived. The preferred answer is accepted as the correct response $r_c$ only if it matches the ground truth label of the original dataset. Any example that does not satisfy this condition is discarded. This procedure ensures that $r_c$ represents a genuinely correct answer in an absolute sense rather than merely a relatively better option within MMPR's preference pairs.

After this filtering step, we retain the image, the instruction $x$, and the validated correct answer $r_c$ as the foundation for generating perceptual and reasoning targeted perturbations. The complete prompt templates used throughout the following steps are provided in Figure S5.

**Step 2. Extraction of perceptual attributes.** Given $(x, r_c)$, we query a strong vision language model (e.g., GPT-5) to extract a small set of perceptual attributes that are both visually grounded, relevant to the instruction, and not explicitly stated in the text instruction itself, ensuring that they reflect information genuinely observed in the image. Concretely, we prompt the model to list at most six atomic attributes such as object counts, shapes, colors, spatial relations, or the presence of specific entities. An example prompt is shown in Figure S6.

We discard attributes that are purely semantic or speculative and keep only those that can be validated visually. In practice, the retained attributes cover counts of people or objects, geometric shapes, colors and textures, relative positions, and the presence or absence of salient elements such as text, axes, or legends in charts.

Dataset Generation Step 2:
EXTRACT PERCEPTION PROMPT

```
Extract 3-6 key perceptual attributes that are
relevant to answering the question and not
explicitly mentioned in the text instruction.
Attributes must reflect information directly
observed in the image, such as object counts,
shapes, colors, spatial relations, or specific
entities.

Examples: counts ("6 objects"), colors
("cyan cylinder"), sizes ("small cube"),
text/numbers ("19")

QUESTION: { question }
ANSWER_GT: { answer_gt }
CHOSEN: { chosen }

Output ONLY this JSON (no markdown, no extra
text): {{"perception_attrs":
["attr1","attr2","attr3"]}}
```

*Figure S6.* The prompt template is used for extracting perceptual attributes. The MLLM extractor outputs visual information relevant to the question but not explicitly mentioned in the instruction.

**Step 3. Generation of a perceptual perturbation (Rejected 1)** Given the perceptual attributes extracted in Step 2, we generate the first negative candidate $r_{r_p}$ by applying a minimal but visually meaningful modification to the attribute set while preserving the structure and style of the correct answer $r_c$.

The goal is to produce an answer that appears plausible without close inspection, yet becomes clearly incorrect once the image is examined. To accomplish this, the model receives both the chosen answer and the attribute list and is instructed to alter one or a small subset of attributes that can be visually validated. Typical edits include changing an object count, altering a color, modifying a spatial relation, or toggling the presence of a specific entity.

Only perturbations that remain visually grounded and internally coherent are accepted. We additionally filter out any perturbed candidate that does not diverge from the ground truth, ensuring that the final $r_{r_p}$ constitutes a genuinely rejected response. A simplified example of the prompt is provided in Figure S7.

**Step 4. Joint perceptual and reasoning perturbation (Rejected 2)** Building on the perceptual attributes identified in Step 2, we generate the second negative candidate $r_{r_{p+r}}$ by introducing a coupled error that affects both perception and reasoning. The model receives the instruction, the correct answer $r_c$, and the validated perceptual attribute list, and is asked to modify one or more visually grounded attributes while additionally inserting a reasoning mistake that depends on these altered perceptual cues.

Dataset Generation Step 3 & 4:
REWRITE REJECTED PROMPT

```
You are given the following information:

QUESTION: { question }
ANSWER_GT: { answer_gt }
CHOSEN: { chosen }
PERCEPTION_ATTRS: { perception_attrs_json }

Based on this information, generate two perturbed
incorrect versions derived from CHOSEN.

CRITICAL: Each perturbed version MUST follow the
EXACT same format as CHOSEN, including:
- Same structure (step-by-step reasoning if
- present)
- Same answer format (e.g., if CHOSEN ends with
{ answer }, your version must too)
- Same level of detail

Perturbation rules:
1. Type 1: Keep the logical structure of the
CHOSEN answer, but slightly alter perception-
related details (as PERCEPTION_ATTRS) so they
contradict the image.
2. Type 2: Alter perception-related details AND
introduce logical errors in the reasoning process.

Each perturbation MUST:
- Follow CHOSEN's exact format
- Include the complete reasoning with modified
- details
- Result in a DIFFERENT final answer than
- ANSWER_GT

Output ONLY this JSON (no markdown, no extra
text):
{{
  "rejected_1": "<full perturbed text for type 1>",
  "rejected_2": "<full perturbed text for type 2>"
}}
```

*Figure S7.* The prompt template was used to generate rejected responses. The generation MLLM produces a perceptual-perturbation response (`"rejected_1"`., $r_{r_p}$) by modifying a small subset of visual attributes in the prompt. A joint-perturbation response (`"rejected_2"`, $r_{r_{p+r}}$) is generated by inserting a reasoning error based on the altered perceptual attributes.

The reasoning error may manifest as an incorrect comparison, a wrong causal interpretation, a misread trend in a chart, or a faulty high-level conclusion that becomes inconsistent once the image is examined. The objective is to produce a fluent response whose logic chain reflects an incorrect perceptual premise and an erroneous inference derived from it.

As with the previous step, we enforce an automatic filtering stage that discards any generated response whose final answer does not diverge from the ground truth. This ensures that the retained $r_{r_{p+r}}$ constitutes a genuinely rejected candidate that simultaneously violates visual facts and reasoning validity. A simplified example of the prompt is provided in Figure S7.

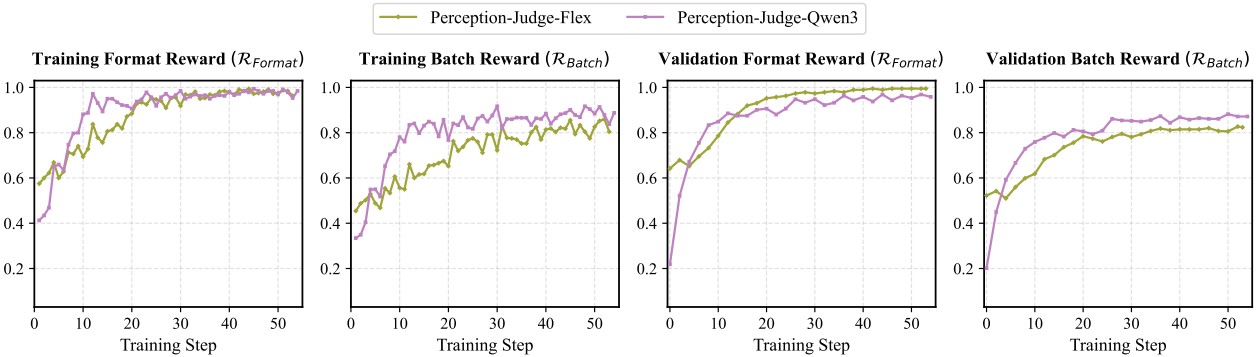

*Figure S8.* Format reward ($\mathcal{R}_{\mathtt{Format}}$) and batch ranking reward ($\mathcal{R}_{\mathtt{Batch}}$) for Perception-Judge-Flex-7B and Perception-Judge-Qwen3-4B on the training and validation splits of PPJD.

## E. Reward Analysis of Perception-Judge

As shown in Figure S8, We analyze the training dynamics of both Perception-Judge-Flex-7B and Perception-Judge-Qwen3-4B by tracking the format reward ($\mathcal{R}_{\mathtt{Format}}$) and batch ranking reward ($\mathcal{R}_{\mathtt{Batch}}$) on training and validation sets. First, regarding structural alignment, both models rapidly converge to near-perfect $\mathcal{R}_{\mathtt{Format}}$ in the early stages, indicating that the syntactic constraints of the output format are easily learnable regardless of the backbone architecture. Second, for $\mathcal{R}_{\mathtt{Batch}}$, which reflects the core ranking capability, both models demonstrate a steady logarithmic improvement before entering a refinement phase. Notably, Perception-Judge-Qwen3 achieves a consistently higher reward trajectory compared to the Flex variant, suggesting that its stronger reasoning priors enables more effective capture of subtle preference signals. Furthermore, the validation curves closely track the training trajectories with a negligible gap across both metrics, confirming that our optimization strategy generalizes robustly without significant overfitting.

## F. Additional Quantitative Evaluation

### F.1. Generalization to a larger model scale

We evaluate whether the proposed training methodology transfers effectively to larger backbones by scaling the judge models beyond the configurations used in the main paper. Specifically, we extend our experiments from Flex-Judge-7B and Qwen3-VL-4B-Thinking to their larger counterparts: Flex-Judge-VL-32B and Qwen3-VL-8B-Thinking. For the Flex-Judge-32B, which applies LoRA (Hu et al., 2022) based fine-tuning to Qwen2.5-VL-32B-Instruct, we train an additional LoRA adapter of rank 32 while keeping the original Flex Judge LoRA modules fixed, yielding Perception-Judge-Flex-32B. For Qwen3-VL-8B-Thinking, we employ full fine-tuning to update all parameters, allowing for more comprehensive adaptation of its reasoning capabilities. Although our main experiments focus on smaller

backbones (7B and 4B), these scale-up studies demonstrate consistent performance gains on both 32B and 8B scales, indicating that the proposed approach generalizes robustly across model sizes and architectures. Detailed results are summarized in Table S2.

### F.2. Score Evaluation on Spearman

We assess the alignment between model-assigned scores and human judgments using Spearman's rank correlation. The proposed Perception-Judge achieves substantially higher correlation compared to baseline multimodal judges, indicating stronger monotonic agreement with human-annotated quality rankings. This improvement reflects the model's ability to suppress visually inconsistent responses and maintain coherent scoring across diverse perceptual tasks. Detailed numerical comparisons are provided in Table S3.

### F.3. Pair Evaluation on F1 and Recall

Pairwise evaluation examines how accurately each judge model identifies the human preferred response in direct comparisons. Perception-Judge surpasses Qwen2.5-VL-7B and Flex-Judge-VL-7B in both recall and F1, demonstrating improved sensitivity to visually grounded correctness and reduced susceptibility to logically fluent but perceptually incorrect outputs. Full results for all datasets and settings can be found in Table S3.

### F.4. Batch Evaluation on Edit Distance

Batch-level ranking metrics further quantify global consistency with human preference orderings. Perception-Judge achieves a lower edit distance and higher Kendall's tau and NDCG scores, indicating more stable ranking structures that better reflect perceptual fidelity. These outcomes confirm the benefits of perceptual grounding in large-scale evaluation scenarios. Comprehensive metric summaries are presented in Table S3.

*Table S2.* Evaluation of 32B scale models on all datasets across score-based, pairwise, and batch-level metrics. ‡: trained with LoRA (Hu et al., 2022). Each judgment is sampled three times and averaged to reduce variance.

| | Model | Size | COCO | C.C. | Diff. | Graphics | Math | Text | WIT | Chart | VisIT | CC-3M | M2W | SciQA | Aes | MM-Vet | Avg. |
|---|---|---|---|---|---|---|---|---|---|---|---|---|---|---|---|---|---|
| **Score (↑)** | GPT-4V † | - | 0.410 | 0.444 | 0.361 | 0.449 | 0.486 | 0.506 | 0.457 | 0.585 | 0.554 | 0.266 | 0.267 | 0.315 | 0.472 | 0.367 | 0.424 |
| | GPT-4o † | - | 0.396 | 0.452 | 0.341 | 0.464 | 0.460 | 0.564 | 0.408 | 0.573 | 0.589 | 0.305 | 0.262 | 0.569 | 0.421 | 0.342 | 0.439 |
| | Gemini-1.0-Pro-Vision * | - | 0.262 | 0.408 | - | 0.400 | 0.228 | 0.222 | 0.418 | 0.343 | 0.336 | 0.374 | 0.324 | 0.073 | 0.360 | 0.207 | 0.304 |
| | Prometheus-V † | 13B | 0.289 | 0.342 | 0.106 | 0.172 | 0.182 | 0.214 | 0.209 | 0.224 | 0.226 | 0.228 | 0.089 | 0.174 | 0.368 | 0.157 | 0.213 |
| | LLaVA-1.5 * | 13B | 0.247 | 0.227 | 0.060 | 0.242 | 0.093 | 0.245 | 0.109 | 0.237 | 0.177 | 0.071 | 0.424 | 0.279 | 0.414 | 0.322 | 0.225 |
| | LLaVA-1.6 * | 34B | 0.285 | 0.251 | -0.012 | 0.262 | 0.238 | 0.258 | 0.151 | 0.318 | 0.198 | 0.109 | 0.022 | 0.206 | 0.025 | 0.265 | 0.184 |
| | LLaVA-Critic † | 7B | 0.382 | 0.450 | 0.103 | 0.316 | 0.356 | 0.378 | 0.179 | 0.421 | 0.322 | 0.246 | 0.301 | 0.269 | 0.395 | 0.272 | 0.314 |
| | Qwen2.5-VL-Instruct | 7B | 0.294 | 0.247 | -0.020 | -0.041 | 0.095 | 0.170 | 0.056 | 0.011 | 0.328 | 0.178 | 0.255 | 0.311 | 0.327 | 0.103 | 0.165 |
| | Flex-Judge-VL | 7B | 0.201 | 0.297 | 0.049 | 0.301 | 0.468 | 0.437 | 0.111 | 0.513 | 0.381 | 0.189 | 0.321 | 0.361 | 0.296 | 0.311 | 0.404 |
| | Flex-Judge-VL ‡ | 32B | 0.234 | 0.337 | 0.215 | 0.322 | 0.343 | 0.372 | 0.171 | 0.528 | 0.434 | 0.195 | 0.115 | 0.425 | 0.377 | 0.33 | 0.396 |
| | Qwen3-VL-Thinking | 4B | 0.342 | 0.392 | 0.271 | 0.405 | 0.353 | 0.489 | 0.284 | 0.607 | 0.388 | 0.225 | 0.074 | 0.521 | 0.319 | 0.308 | 0.419 |
| | Qwen3-VL-Thinking | 8B | 0.324 | 0.415 | 0.277 | 0.4 | 0.451 | 0.46 | 0.332 | 0.587 | 0.414 | 0.261 | 0.106 | 0.504 | 0.295 | 0.352 | 0.436 |
| | Perception-Judge-Flex | 7B | 0.293 | 0.277 | 0.14 | 0.358 | 0.487 | 0.429 | 0.164 | 0.545 | 0.447 | 0.225 | 0.366 | 0.399 | 0.401 | 0.374 | 0.466 |
| | Perception-Judge-Flex ‡ | 32B | 0.376 | 0.462 | 0.195 | 0.395 | 0.469 | 0.434 | 0.161 | 0.566 | 0.449 | 0.229 | 0.147 | 0.462 | 0.435 | 0.310 | 0.447 |
| | Perception-Judge-Qwen3 | 4B | 0.348 | 0.410 | 0.347 | 0.401 | 0.401 | 0.489 | 0.225 | 0.609 | 0.462 | 0.273 | 0.142 | 0.556 | 0.344 | 0.332 | 0.457 |
| | Perception-Judge-Qwen3 | 8B | 0.377 | 0.462 | 0.314 | 0.407 | 0.481 | 0.450 | 0.362 | 0.598 | 0.450 | 0.275 | 0.115 | 0.544 | 0.348 | 0.270 | 0.459 |
| **Pair w. Tie (↑)** | GPT-4V † | - | 0.539 | 0.634 | 0.668 | 0.632 | 0.459 | 0.495 | 0.536 | 0.369 | 0.591 | 0.544 | 0.544 | 0.389 | 0.620 | 0.517 | 0.538 |
| | GPT-4o † | - | 0.539 | 0.634 | 0.668 | 0.632 | 0.459 | 0.495 | 0.536 | 0.369 | 0.591 | 0.544 | 0.544 | 0.389 | 0.620 | 0.517 | 0.538 |
| | Gemini-1.0-Pro-Vision * | - | 0.616 | 0.787 | - | 0.650 | 0.436 | 0.664 | 0.605 | 0.500 | 0.660 | 0.560 | 0.370 | 0.262 | 0.190 | 0.312 | 0.509 |
| | LLaVA-1.5 * | 13B | 0.273 | 0.478 | 0.286 | 0.273 | 0.657 | 0.510 | 0.369 | 0.383 | 0.456 | 0.484 | 0.347 | 0.223 | 0.389 | 0.254 | 0.384 |
| | LLaVA-Critic † | 7B | 0.593 | 0.687 | 0.707 | 0.587 | 0.432 | 0.544 | 0.564 | 0.338 | 0.596 | 0.628 | 0.591 | 0.370 | 0.686 | 0.464 | 0.556 |
| | Qwen2.5-VL-Instruct | 7B | 0.446 | 0.474 | 0.507 | 0.326 | 0.397 | 0.383 | 0.366 | 0.364 | 0.461 | 0.483 | 0.358 | 0.442 | 0.494 | 0.420 | 0.423 |
| | Flex-Judge-VL | 7B | 0.504 | 0.624 | 0.611 | 0.552 | 0.454 | 0.518 | 0.413 | 0.403 | 0.565 | 0.562 | 0.588 | 0.389 | 0.589 | 0.507 | 0.514 |
| | Flex-Judge-VL ‡ | 32B | 0.586 | 0.685 | 0.868 | 0.597 | 0.481 | 0.528 | 0.601 | 0.374 | 0.615 | 0.634 | 0.625 | 0.386 | 0.677 | 0.529 | 0.57 |
| | Qwen3-VL-Thinking | 4B | 0.547 | 0.653 | 0.743 | 0.559 | 0.48 | 0.495 | 0.613 | 0.367 | 0.586 | 0.595 | 0.593 | 0.347 | 0.666 | 0.493 | 0.543 |
| | Qwen3-VL-Thinking | 8B | 0.56 | 0.661 | 0.699 | 0.579 | 0.481 | 0.51 | 0.591 | 0.357 | 0.589 | 0.604 | 0.604 | 0.343 | 0.669 | 0.512 | 0.546 |
| | Perception-Judge-Flex | 7B | 0.541 | 0.671 | 0.567 | 0.562 | 0.454 | 0.511 | 0.444 | 0.384 | 0.577 | 0.592 | 0.554 | 0.367 | 0.629 | 0.482 | 0.520 |
| | Perception-Judge-Flex ‡ | 32B | 0.589 | 0.686 | 0.852 | 0.625 | 0.502 | 0.534 | 0.612 | 0.367 | 0.607 | 0.625 | 0.633 | 0.370 | 0.683 | 0.498 | 0.573 |
| | Perception-Judge-Qwen3 | 4B | 0.541 | 0.661 | 0.777 | 0.593 | 0.489 | 0.510 | 0.605 | 0.369 | 0.583 | 0.604 | 0.613 | 0.352 | 0.695 | 0.504 | 0.554 |
| | Perception-Judge-Qwen3 | 8B | 0.567 | 0.663 | 0.744 | 0.601 | 0.486 | 0.529 | 0.604 | 0.379 | 0.607 | 0.602 | 0.613 | 0.341 | 0.698 | 0.514 | 0.558 |
| **Pair w.o. Tie (↑)** | GPT-4V † | - | 0.729 | 0.772 | 0.884 | 0.853 | 0.665 | 0.661 | 0.760 | 0.495 | 0.785 | 0.707 | 0.697 | 0.639 | 0.741 | 0.654 | 0.717 |
| | GPT-4o † | - | 0.774 | 0.776 | 0.934 | 0.835 | 0.628 | 0.618 | 0.737 | 0.513 | 0.741 | 0.770 | 0.706 | 0.722 | 0.887 | 0.660 | 0.736 |
| | Gemini-1.0-Pro-Vision * | - | 0.717 | 0.840 | - | 0.770 | 0.678 | 0.793 | 0.688 | 0.658 | 0.711 | 0.652 | 0.471 | 0.358 | 0.265 | 0.400 | 0.615 |
| | LLaVA-1.5 * | 13B | 0.327 | 0.537 | 0.302 | 0.300 | 0.726 | 0.684 | 0.600 | 0.610 | 0.648 | 0.583 | 0.449 | 0.443 | 0.498 | 0.344 | 0.504 |
| | LLaVA-Critic † | 7B | 0.771 | 0.774 | 0.755 | 0.758 | 0.596 | 0.658 | 0.680 | 0.488 | 0.727 | 0.742 | 0.692 | 0.658 | 0.715 | 0.635 | 0.689 |
| | Qwen2.5-VL-Instruct | 7B | 0.479 | 0.492 | 0.510 | 0.268 | 0.368 | 0.394 | 0.334 | 0.348 | 0.506 | 0.538 | 0.330 | 0.511 | 0.486 | 0.388 | 0.425 |
| | Flex-Judge-VL | 7B | 0.644 | 0.691 | 0.634 | 0.693 | 0.573 | 0.595 | 0.491 | 0.564 | 0.667 | 0.656 | 0.651 | 0.660 | 0.600 | 0.630 | 0.623 |
| | Flex-Judge-VL ‡ | 32B | 0.761 | 0.764 | 0.906 | 0.744 | 0.588 | 0.614 | 0.715 | 0.501 | 0.738 | 0.743 | 0.694 | 0.686 | 0.693 | 0.677 | 0.694 |
| | Qwen3-VL-Thinking | 4B | 0.708 | 0.726 | 0.776 | 0.696 | 0.604 | 0.567 | 0.724 | 0.513 | 0.699 | 0.697 | 0.66 | 0.606 | 0.682 | 0.656 | 0.663 |
| | Qwen3-VL-Thinking | 8B | 0.725 | 0.734 | 0.733 | 0.719 | 0.599 | 0.59 | 0.703 | 0.501 | 0.703 | 0.704 | 0.676 | 0.618 | 0.687 | 0.712 | 0.669 |
| | Perception-Judge-Flex | 7B | 0.703 | 0.746 | 0.592 | 0.715 | 0.604 | 0.594 | 0.532 | 0.546 | 0.697 | 0.694 | 0.622 | 0.665 | 0.646 | 0.652 | 0.645 |
| | Perception-Judge-Flex ‡ | 32B | 0.769 | 0.766 | 0.894 | 0.789 | 0.636 | 0.626 | 0.732 | 0.520 | 0.732 | 0.733 | 0.712 | 0.690 | 0.702 | 0.686 | 0.709 |
| | Perception-Judge-Qwen3 | 4B | 0.707 | 0.737 | 0.815 | 0.749 | 0.649 | 0.597 | 0.726 | 0.532 | 0.706 | 0.708 | 0.689 | 0.663 | 0.716 | 0.717 | 0.691 |
| | Perception-Judge-Qwen3 | 8B | 0.741 | 0.740 | 0.781 | 0.761 | 0.652 | 0.620 | 0.725 | 0.545 | 0.734 | 0.705 | 0.688 | 0.642 | 0.719 | 0.747 | 0.698 |
| **Batch (↓)** | GPT-4V † | - | 0.318 | 0.353 | 0.070 | 0.385 | 0.348 | 0.319 | 0.290 | 0.347 | 0.300 | 0.402 | 0.597 | 0.462 | 0.453 | 0.411 | 0.361 |
| | Gemini-1.0-Pro-Vision * | - | 0.287 | 0.299 | - | 0.473 | 0.462 | 0.430 | 0.344 | 0.520 | 0.426 | 0.357 | 0.613 | 0.412 | 0.467 | 0.529 | 0.432 |
| | LLaVA-1.5 * | 13B | 0.577 | 0.492 | 0.562 | 0.535 | 0.598 | 0.650 | 0.616 | 0.644 | 0.620 | 0.563 | 0.639 | 0.563 | 0.650 | 0.652 | 0.597 |
| | Qwen2.5-VL-Instruct | 7B | 0.562 | 0.450 | 0.593 | 0.630 | 0.607 | 0.582 | 0.631 | 0.570 | 0.569 | 0.519 | 0.639 | 0.703 | 0.558 | 0.572 | 0.585 |
| | Flex-Judge-VL | 7B | 0.487 | 0.462 | 0.531 | 0.559 | 0.505 | 0.543 | 0.576 | 0.482 | 0.486 | 0.537 | 0.526 | 0.551 | 0.521 | 0.478 | 0.517 |
| | Flex-Judge-VL ‡ | 32B | 0.45 | 0.435 | 0.361 | 0.516 | 0.488 | 0.476 | 0.499 | 0.452 | 0.464 | 0.487 | 0.524 | 0.46 | 0.49 | 0.439 | 0.469 |
| | Qwen3-VL-Thinking | 4B | 0.532 | 0.46 | 0.401 | 0.535 | 0.514 | 0.513 | 0.507 | 0.452 | 0.544 | 0.539 | 0.534 | 0.453 | 0.488 | 0.47 | 0.498 |
| | Qwen3-VL-Thinking | 8B | 0.461 | 0.415 | 0.339 | 0.471 | 0.502 | 0.456 | 0.466 | 0.43 | 0.488 | 0.514 | 0.555 | 0.452 | 0.511 | 0.47 | 0.469 |
| | Perception-Judge-Flex | 7B | 0.467 | 0.428 | 0.487 | 0.535 | 0.492 | 0.516 | 0.547 | 0.529 | 0.456 | 0.528 | 0.540 | 0.560 | 0.541 | 0.435 | 0.505 |
| | Perception-Judge-Flex ‡ | 32B | 0.438 | 0.392 | 0.341 | 0.504 | 0.439 | 0.439 | 0.451 | 0.431 | 0.434 | 0.448 | 0.559 | 0.474 | 0.506 | 0.439 | 0.449 |
| | Perception-Judge-Qwen3 | 4B | 0.447 | 0.371 | 0.296 | 0.491 | 0.448 | 0.436 | 0.415 | 0.426 | 0.432 | 0.521 | 0.578 | 0.431 | 0.479 | 0.434 | 0.444 |
| | Perception-Judge-Qwen3 | 8B | 0.421 | 0.378 | 0.293 | 0.463 | 0.446 | 0.421 | 0.372 | 0.406 | 0.433 | 0.483 | 0.550 | 0.438 | 0.473 | 0.435 | 0.431 |

*Table S3.* Additional quantitative evaluation of 7B scale models on complementary metrics, including Spearman correlation, F1 score, Recall, Kendall's tau, and NDCG.

| | Model | Size | COCO | C.C. | Diff. | Graphics | Math | Text | WIT | Chart | VisIT | CC-3M | M2W | SciQA | Aes | MM-Vet | Ave. |
|---|---|---|---|---|---|---|---|---|---|---|---|---|---|---|---|---|---|
| **Score (↑)** | *Spearman* | | | | | | | | | | | | | | | | |
| | Qwen2.5-VL-Instruct | 7B | 0.317 | 0.176 | 0.114 | 0.292 | 0.323 | 0.323 | 0.238 | 0.521 | 0.378 | 0.201 | 0.192 | 0.252 | 0.339 | 0.172 | 0.274 |
| | Flex-Judge-VL | 7B | 0.258 | 0.333 | 0.007 | 0.379 | 0.471 | 0.421 | 0.039 | 0.574 | 0.333 | 0.256 | 0.062 | 0.324 | 0.265 | 0.300 | 0.287 |
| | Perception-Judge-Flex | 7B | 0.367 | 0.324 | 0.033 | 0.403 | 0.476 | 0.435 | 0.113 | 0.579 | 0.377 | 0.264 | 0.162 | 0.395 | 0.261 | 0.287 | **0.320** |
| **Pair w.o. Tie (↑)** | *F1-score* | | | | | | | | | | | | | | | | |
| | Qwen2.5-VL-Instruct | 7B | 0.638 | 0.682 | 0.401 | 0.575 | 0.474 | 0.553 | 0.514 | 0.519 | 0.648 | 0.694 | 0.633 | 0.602 | 0.617 | 0.517 | 0.576 |
| | Flex-Judge-VL | 7B | 0.648 | 0.693 | 0.424 | 0.645 | 0.559 | 0.619 | 0.502 | 0.576 | 0.680 | 0.657 | 0.652 | 0.648 | 0.624 | 0.570 | 0.607 |
| | Perception-Judge-Flex | 7B | 0.693 | 0.733 | 0.386 | 0.684 | 0.566 | 0.618 | 0.561 | 0.560 | 0.687 | 0.709 | 0.663 | 0.678 | 0.644 | 0.602 | **0.627** |
| **Pair w.o. Tie (↑)** | *Recall* | | | | | | | | | | | | | | | | |
| | Qwen2.5-VL-Instruct | 7B | 0.585 | 0.647 | 0.408 | 0.569 | 0.417 | 0.498 | 0.461 | 0.541 | 0.619 | 0.645 | 0.571 | 0.540 | 0.517 | 0.466 | 0.535 |
| | Flex-Judge-VL | 7B | 0.651 | 0.710 | 0.435 | 0.663 | 0.540 | 0.605 | 0.496 | 0.630 | 0.679 | 0.663 | 0.631 | 0.625 | 0.593 | 0.550 | 0.605 |
| | Perception-Judge-Flex | 7B | 0.705 | 0.743 | 0.452 | 0.713 | 0.564 | 0.619 | 0.560 | 0.644 | 0.692 | 0.709 | 0.674 | 0.679 | 0.648 | 0.622 | **0.645** |
| **Batch (↑)** | *Kendall's Tau* | | | | | | | | | | | | | | | | |
| | Qwen2.5-VL-Instruct | 7B | 0.233 | 0.500 | 0.000 | 0.153 | 0.217 | 0.246 | 0.316 | 0.256 | 0.383 | 0.333 | 0.068 | 0.312 | 0.320 | 0.258 | 0.257 |
| | Flex-Judge-VL | 7B | 0.327 | 0.450 | 0.080 | 0.253 | 0.365 | 0.323 | 0.187 | 0.387 | 0.447 | 0.237 | 0.020 | 0.035 | 0.251 | 0.357 | 0.266 |
| | Perception-Judge-Flex | 7B | 0.410 | 0.467 | 0.137 | 0.217 | 0.378 | 0.297 | 0.247 | 0.307 | 0.447 | 0.270 | 0.073 | 0.139 | 0.271 | 0.469 | **0.295** |
| **Batch (↑)** | *NDCG* | | | | | | | | | | | | | | | | |
| | Qwen2.5-VL-Instruct | 7B | 0.910 | 0.845 | 0.853 | 0.842 | 0.856 | 0.853 | 0.869 | 0.851 | 0.894 | 0.878 | 0.838 | 0.879 | 0.898 | 0.873 | 0.867 |
| | Flex-Judge-VL | 7B | 0.880 | 0.879 | 0.875 | 0.871 | 0.879 | 0.874 | 0.843 | 0.890 | 0.900 | 0.862 | 0.805 | 0.836 | 0.847 | 0.883 | 0.866 |
| | Perception-Judge-Flex | 7B | 0.901 | 0.915 | 0.883 | 0.858 | 0.888 | 0.872 | 0.85 | 0.856 | 0.899 | 0.864 | 0.821 | 0.844 | 0.856 | 0.900 | **0.872** |

# G. Additional Qualitative Results

## G.1. MLLM-as-a-Judge Qualitative Results

This section presents additional qualitative evidence Figure S9, Figure S12, Figure S13, and Figure S14 illustrating how the proposed Perception-Judge corrects systematic perceptual failures observed in baseline multimodal judges such as Qwen2.5-VL-7B (Bai et al., 2023), Flex-Judge-VL-7B (Ko et al., 2025), and Qwen3-VL-4B-Thinking (Yang et al., 2025). The examples highlight two recurring error modes in these models: inaccurate visual interpretation and over-reliance on response-anchored cues.

Across various cases, baseline judges assign high scores to responses that are logically fluent yet perceptually incorrect. In contrast, Perception-Judge consistently identifies discrepancies between the visual input and the candidate responses, penalizing misaligned descriptions and favoring perceptually grounded reasoning. The qualitative cases provided in the following pages serve as direct references demonstrating this improvement, particularly in scenarios involving miscounted objects, incorrect color or spatial attributes, or hallucinated visual elements.

Collectively, these examples reinforce that Perception-Judge more reliably aligns evaluation with the underlying visual evidence and mitigates cognitive biases prevalent in existing MLLM-as-a-Judge systems.

# H. Additional Ablation Study

## H.1. Detailed Results of Ablation Study

Due to space limitations in the main paper, we provide the full ablation results. These results present a comprehensive analysis of how dataset composition and reward design influence the performance of Perception-Judge.

From a dataset perspective, PPJD demonstrates clear and consistent improvements over the MMPR baseline across all evaluated benchmarks. The gains are especially pronounced on datasets that demand precise perceptual grounding, where MMPR frequently fails to prevent visually inconsistent scoring. This indicates that PPJD's perceptually aligned data more effectively supports robust judgment behavior.

From a reward perspective, the comparison between pair reward and batch reward shows that the proposed **batch reward** provides more stable and reliable supervision. While the pair reward offers a minimal and local comparison signal that encourages short-range perceptual preference alignment, it remains limited in its ability to enforce global consistency. In contrast, the batch reward introduces structured ranking information that leads to better alignment across score-based, pairwise, and ranking-oriented metrics. For completeness, we include a concise description of the **pairwise reward** mechanism below.

Complete quantitative outcomes for all datasets and reward configurations are summarized in Table S4.

**Pairwise Reward.** To guide the judge toward perceptu-

*Table S4.* Full ablation results comparing MMPR and PPJD across all datasets and reward settings. The table reports performance under score-based, pairwise, and batch-level metrics. From a dataset perspective, PPJD consistently improves over the MMPR baseline across diverse benchmarks. From a reward perspective, the proposed batch reward yields more stable and globally aligned supervision than the pair reward.

| | Dataset | Reward | COCO | C.C. | Diff. | Graphics | Math | Text | WIT | Chart | VisIT | CC-3M | M2W | SciQA | Aes | MM-Vet | Ave. |
|---|---|---|---|---|---|---|---|---|---|---|---|---|---|---|---|---|---|
| **Score (↑)** | - | - | 0.201 | 0.297 | 0.049 | 0.301 | 0.468 | 0.437 | 0.111 | 0.513 | 0.381 | 0.189 | 0.321 | 0.361 | 0.296 | 0.311 | 0.404 |
| | MMPR | Pair | 0.301 | 0.292 | 0.048 | 0.375 | 0.464 | 0.413 | 0.129 | 0.552 | 0.448 | 0.219 | 0.299 | 0.376 | 0.384 | 0.354 | 0.454 |
| | PPJD (Ours) | Pair | 0.283 | 0.220 | 0.077 | 0.376 | 0.479 | 0.419 | 0.104 | 0.559 | 0.441 | 0.217 | 0.287 | 0.399 | 0.355 | 0.382 | 0.458 |
| | PPJD (Ours) | Batch | 0.342 | 0.285 | 0.106 | 0.392 | 0.484 | 0.451 | 0.165 | 0.576 | 0.456 | 0.269 | 0.307 | 0.396 | 0.389 | 0.409 | **0.476** |
| **Pair w. Tie (↑)** | - | - | 0.504 | 0.624 | 0.611 | 0.552 | 0.454 | 0.518 | 0.413 | 0.403 | 0.565 | 0.562 | 0.588 | 0.389 | 0.589 | 0.507 | 0.514 |
| | MMPR | Pair | 0.536 | 0.659 | 0.529 | 0.570 | 0.454 | 0.517 | 0.434 | 0.389 | 0.576 | 0.584 | 0.546 | 0.366 | 0.627 | 0.449 | 0.515 |
| | PPJD (Ours) | Pair | 0.543 | 0.661 | 0.508 | 0.571 | 0.460 | 0.523 | 0.449 | 0.369 | 0.582 | 0.606 | 0.530 | 0.359 | 0.644 | 0.457 | **0.518** |
| | PPJD (Ours) | Batch | 0.553 | 0.690 | 0.471 | 0.575 | 0.458 | 0.514 | 0.456 | 0.381 | 0.585 | 0.618 | 0.520 | 0.356 | 0.634 | 0.442 | **0.518** |
| **Pair w.o. Tie (↑)** | - | - | 0.644 | 0.691 | 0.634 | 0.693 | 0.573 | 0.595 | 0.491 | 0.564 | 0.667 | 0.656 | 0.651 | 0.660 | 0.600 | 0.630 | 0.623 |
| | MMPR | Pair | 0.698 | 0.735 | 0.554 | 0.723 | 0.593 | 0.605 | 0.520 | 0.560 | 0.697 | 0.684 | 0.614 | 0.688 | 0.646 | 0.624 | 0.641 |
| | PPJD (Ours) | Pair | 0.708 | 0.738 | 0.533 | 0.725 | 0.607 | 0.612 | 0.537 | 0.528 | 0.703 | 0.711 | 0.596 | 0.665 | 0.663 | 0.628 | 0.644 |
| | PPJD (Ours) | Batch | 0.722 | 0.769 | 0.494 | 0.731 | 0.610 | 0.600 | 0.547 | 0.548 | 0.708 | 0.724 | 0.584 | 0.680 | 0.653 | 0.637 | **0.648** |
| **Batch (↓)** | - | - | 0.487 | 0.462 | 0.531 | 0.559 | 0.505 | 0.543 | 0.576 | 0.482 | 0.486 | 0.537 | 0.526 | 0.551 | 0.521 | 0.478 | 0.517 |
| | MMPR | Pair | 0.488 | 0.433 | 0.516 | 0.566 | 0.521 | 0.531 | 0.55 | 0.512 | 0.453 | 0.514 | 0.546 | 0.582 | 0.526 | 0.455 | 0.515 |
| | PPJD (Ours) | Pair | 0.491 | 0.461 | 0.467 | 0.556 | 0.509 | 0.536 | 0.554 | 0.499 | 0.466 | 0.522 | 0.548 | 0.575 | 0.524 | 0.462 | 0.513 |
| | PPJD (Ours) | Batch | 0.463 | 0.425 | 0.497 | 0.528 | 0.492 | 0.515 | 0.514 | 0.518 | 0.448 | 0.491 | 0.557 | 0.556 | 0.535 | 0.456 | **0.500** |

*Table S5.* Ablation on the KL loss coefficient.

| KL coef | Score (↑) | Pair w. Tie (↑) | Pair w.o. Tie (↑) | Batch (↓) |
|---|---|---|---|---|
| 5e-2 | 0.450 | 0.517 | 0.639 | 0.506 |
| 1e-2 | **0.466** | **0.520** | **0.645** | 0.505 |
| 5e-3 | 0.457 | 0.519 | **0.645** | **0.503** |
| 1e-3 | 0.457 | 0.518 | 0.642 | 0.504 |

ally grounded decisions, we provide a minimal comparison signal between responses of increasing perceptual degradation. This pairwise feedback encourages local pairwise consistency and focuses optimization on perceptual preference alignment rather than absolute scoring. We enforce two target preferences, $r_c \succ r_{r_p}$ and $r_{r_p} \succ r_{r_{p+r}}$. For each instance, a comparison pair $(a, b)$ is randomly sampled with equal probability from $\{(r_c, r_{r_p}), (r_{r_p}, r_{r_{p+r}})\}$. Let $\widehat{P}_i(a \succ b) \in \{0, 1\}$ indicate whether the judge's pairwise decision matches the target preference. The pairwise reward is defined as:

$$\mathcal{R}_{\text{Pair}}(o_i) = \widehat{P}_i(a \succ b). \tag{S3}$$

This stochastic formulation promotes consistent preference learning across perceptual perturbation levels.

### H.2. Ablation studies on KL penalty

The KL coefficient serves as a critical regularizer in GRPO, preventing the model from deviating excessively from the reference policy. Table S5 summarizes the impact of varying the KL coefficient on Flex-Judge-7B. We observe that the model achieves peak performance with a coefficient of 1e-2, striking an optimal balance between exploration and stability.

## I. Failure Cases

Despite the improvements introduced by our approach, certain limitations remain. First, the judge may still fail on tasks that require extremely fine-grained or atypical perceptual recognition, especially when visual cues are subtle or ambiguous. These challenges become more pronounced in scenarios involving implicit three-dimensional structure or complex spatial relationships, which current multimodal backbones are not yet equipped to fully interpret, as illustrated in Figure S15. Such failures reflect broader perception limits of existing models rather than issues specific to our training objective.

Second, responses that demand complex multi-step reasoning can still lead to incorrect judgments. Although these reasoning-heavy cases are not the primary focus of this work, they highlight the need for further advancements in base model reasoning capability to achieve fully reliable judgment in settings where visual perception alone is insufficient.

### Instruction

Solvent volume: 40 mL     Solvent volume: 40 mL

**Solution A**     **Solution B**

Which solution has a higher concentration of green particles? Here is the hint to the Question: **The diagram below is a model of two solutions. Each green ball represents one particle of solute.**

### Response Model: GPT-4

The image shows two flasks labeled Solution A and Solution B, with both containing green particles and an indicated solvent volume of 40 mL for each. Concentration is defined as the amount of solute (in this case, represented by green particles) per unit volume of solvent.

To determine which solution has a higher concentration of green particles, we need to compare the number of green particles in each solution relative to the same volume of solvent.

In Solution A, there are nine green particles, while in Solution B, there are seven green particles. Since both solutions have the same volume of solvent (40 mL), the solution with the greater number of particles per unit volume will have a higher concentration.
Solution A has nine particles in 40 mL of solvent, while Solution B has seven particles in 40 mL of solvent. Therefore, Solution A has a higher concentration of green particles than Solution B.

### Human

Score: 1 / 5

### Qwen3-4B-Thinking

So, let's check the assistant's answer. First, the question is about concentration of green particles. The hint says each green ball is a solute particle. The image shows Solution A with green particles and Solution B with green particles. Wait, the assistant counted Solution A as nine and Solution B as seven. Let's count the particles in the image.

Looking at Solution A: the left flask. Let's count the green circles. Let's see: top row maybe? Wait, the image for Solution A: how many green circles? Let me count. The user provided the image. Wait, in the description, Solution A has Solvent volume: 40 mL and Solution B same. The assistant's answer says Solution A has nine particles, Solution B has seven. Wait, let's count the particles.

In Solution A (left flask), the green circles: let's count. There are 9 green circles. In Solution B (right flask), there are 7 green circles.

...

Solution B: right flask. Let's count:

- Top row: 4 green circles? Wait, no. Wait, Solution B: maybe 7? Wait, the assistant says Solution B has 7. Wait, but maybe the actual count is different. Wait, the user's image: let's look again. Wait, the problem says each green ball is a solute particle. The assistant's answer states Solution A has 9, Solution B has 7. But wait, maybe the correct count is different.
...

Is there any mistake in the reasoning? The steps are: define concentration as solute per volume, compare numbers since volumes are equal, higher number = higher concentration. Correct.

The count: 9 vs 7. Correct.

So the answer is accurate. Therefore, score is high.</think>

**Score: 5**

### Perception-Judge-Qwen3-4B

So, let's evaluate the assistant's answer. First, check helpfulness: the answer explains the concept of concentration and compares the number of particles. Relevance: it directly addresses the question about which solution has a higher concentration. Accuracy: let's count the particles. Wait, in the image, Solution A has 9 green particles? Wait, let me check again. The user's image: Solution A has 9 green circles? Wait, maybe I miscounted. Wait, the problem says ëach green ball represents one particle of solute.Ļet's count:

...

Solution A (left): let's count the green circles. Top: 2, middle: 3, bottom: 4. 2+3+4=9. Correct.

Solution B (right): top: 3, middle: 3, bottom: 4. 3+3+4=10? Wait, no. Wait, maybe the image shows Solution B with 10 particles. Wait, maybe the assistant made a mistake. Wait, the assistant says Solution B has 7, but actually, let's count again.

...
Ah, so Solution B has 10, Solution A has 9. Therefore, Solution B has higher concentration.

The assistant's answer says Solution A has 9 and Solution B has 7, which is wrong. So the conclusion is wrong.

Therefore, the assistant's answer is incorrect.

So the score would be low, maybe 3 or 4.

Wait, let's see:

The assistant's reasoning is step-by-step, but the count is wrong. So the answer is not accurate.

Therefore, the score is low. </think>

**Score: 1**

*Figure S9.* Additional qualitative example of score setting.

## Instruction

**COVID-19: INDIA'S HEALTHCARE IN ICU?**

Please analyse this figure in detail and answer the following question based on this figure: **What is the number of hospital beds per 1,000 people in Russia?**

### Response Model: Gemini

The figure shows the number of hospital beds per 1,000 people in different countries. Russia has 8.2 hospital beds per 1,000 people. This is lower than the number of hospital beds per 1,000 people in Brazil (12.8), China (4.2), and South Africa (2.8). It is higher than the number of hospital beds per 1,000 people in India (0.7).

### Human

Score: 1 / 5

## Qwen3-4B-Thinking

So, let's tackle this. The user wants to know the number of hospital beds per 1,000 people in Russia based on the figure provided.

First, I need to check the figure. The figure has a section with Ḧospital beds (per 1,000 people)änd a bar chart. Let's recall the data points: the countries listed are India, Brazil, China, South Africa, and Russia. The values for each are: India 0.7, Brazil 12.8, China 4.2, South Africa 2.8, and Russia 8.2. Wait, the assistant's answer says Russia has 8.2, which matches the figure. Let me confirm: the bar chart under Ḧospital beds (per 1,000 people)s̈hows Russia at 8.2. So the assistant correctly identified that. Now, checking for accuracy: the figure's data points are as per the bar chart. The answer is correct. Helpfulness: the assistant clearly states the number and relates it to other countries, which is helpful for context. Relevance: directly answers the question. Accuracy: yes, 8.2 is correct. Level of detail: gives the exact number and a comparison, which adds context but isn't excessive. So this answer seems solid. Let's see if there's any mistake. The question is about Russia, and the figure's bar for Russia is 8.2. Yep, that's right. So the assistant's answer is correct and helpful.</think>

**Score: 5**

## Perception-Judge-Qwen3-4B

So, let's check the figure provided. The user is asking for the number of hospital beds per 1,000 people in Russia. First, I need to locate the section that shows hospital beds per 1,000 people. Looking at the infographic, there's a bar chart labeled Ḧospital beds (per 1,000 people)ẅith countries: Brazil (12.8), China (4.2), South Africa (2.8), and India (0.7). Wait, the assistant's answer mentions Russia as 8.2, but in the figure, the countries listed for hospital beds are Brazil, China, South Africa, and India. Russia isn't mentioned there. Let me check again.

Wait, the infographic has a section Ä Glimpse at Current Healthcare Infrastructureẅith a bar chart. The x-axis is Ḧospital beds (per 1,000 people)änd the countries are Brazil (12.8), China (4.2), South Africa (2.8), and India (0.7). There's no Russia in that bar chart. The assistant's answer incorrectly states Russia has 8.2. Maybe the assistant confused the data. Let's check the other parts: the P̈hysicians (per 1,000 people)s̈ection has Brazil (2.5), China (1.4), South Africa (1.4), India (0.8). But Russia isn't in either of those sections. So the assistant made a mistake by including Russia with 8.2, which isn't present in the figure. Therefore, the answer is incorrect because the figure doesn't have Russia's data for hospital beds per 1,000 people. The correct answer would be that the figure doesn't provide Russia's hospital bed data, but the assistant's answer claims it's 8.2, which is wrong.</think>

**Score: 1**

*Figure S10.* Additional qualitative example of score setting.

**Instruction**

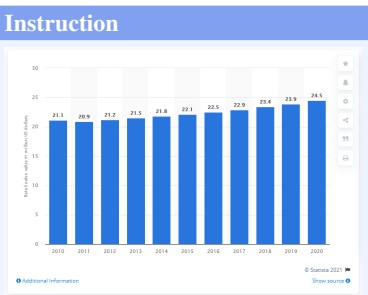

This is a task of reasoning on chart data. Please analyze this figure in detail and answer the following question with reason based on this figure: **What was the retail sales value of organic biscuits and snack bars in 2010?**

**Response Model: Gemini**

The graph shows the retail sales value of organic biscuits and snack bars from 2010 to 2020. The value was 21.1 billion US dollars in 2010 and increased to 24.5 billion US dollars in 2020. So the answer is 21.1 billion US dollars.

**Human**

Score: 5 / 5

**Qwen3-4B-Thinking**

So, let's check the assistant's answer. The question asks for the retail sales value in 2010. The chart is a bar graph with years on the x-axis and retail sales value in million USD on the y-axis. The 2010 bar has a value of 21.1. Wait, the y-axis labels say Retail sales value in million US dollars. So 21.1 is in millions? Wait, the problem is, the assistant wrote 21.1 billion but the chart says million. Let me check the chart description. The y-axis is labeled Retail sales value in million US dollars, so the numbers are in millions. For example, 21.1 on the y-axis is 21.1 million USD. But the assistant said 21.1 billion, which is wrong. Let's confirm: 21.1 million is 0.0211 billion, so the assistant's mistake is the unit. The chart is in millions, so the correct answer is 21.1 million US dollars. The assistant's answer incorrectly stated it as billion, which is a major error. So the answer is wrong because of unit conversion mistake. Let's see the chart: each bar's height corresponds to the value. The 2010 bar is at 21.1 on the y-axis, which is labeled million US dollars, so 21.1 million. The assistant said 21.1 billion, which is wrong. Therefore, the assistant's answer is inaccurate because of unit error. Also, they didn't mention the unit in the answer, but the problem is the unit. So the error is in the numerical value's unit. Therefore, the assistant's answer is incorrect because it misreported the unit (billion instead of million). So the score should be low because of accuracy issue. Let's check the criteria: accuracy is key here. The assistant made a mistake in unit conversion, so accuracy is off.</think>

**Score: 1**

**Perception-Judge-Qwen3-4B**

So, let's see. The user is asking for the retail sales value of organic biscuits and snack bars in 2010 based on the chart. First, I need to check the chart. The x-axis is the years from 2010 to 2020, and the y-axis is the retail sales value in billion US dollars. Looking at the bar for 2010, the height corresponds to 21.1. The assistant's answer says 21.1 billion US dollars, which matches the chart. Let me confirm: the bar for 2010 is labeled 21.1, so that's correct. The answer is accurate, helpful, and directly answers the question. No mistakes here.</think>

**Score: 5**

*Figure S11.* Additional qualitative example of score setting.

### Instruction

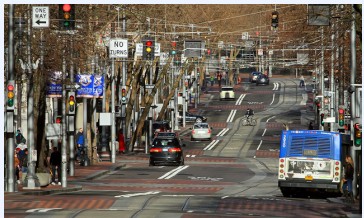

This is a figure including mathematical information. Please analyze this figure in detail and answer the following question with hint based on this figure: Hint: Please answer the question requiring an integer answer and provide the final value, e.g., 1, 2, 3, at the end.

**Question: How many miles per gallon do an average city bus get?**

### Response Model: LLaVA-1.5-13B

Unfortunately, the image does not provide information about the average miles per gallon of a city bus. The image depicts a city street with various vehicles, including cars, a bus, and a person riding a bicycle, but it does not provide any information about the fuel efficiency of these vehicles.

### Human

Score: 4 / 5

### Qwen2.5-VL-7B

The assistant's response is relevant to the user's instruction as it addresses the lack of information in the provided image regarding the fuel efficiency of a city bus. However, the response does not provide the required integer answer to the question. The assistant correctly identifies that the image does not contain the necessary data, which is a fair and accurate observation. The response is clear and concise, but it lacks the specific numerical answer that the user is seeking.

Judgement: 2

### Flex-Judge-VL-7B

Assistant's answer is quite accurate in the sense that the image does not provide any specific data about the fuel efficiency of a city bus. However, the answer could be more direct and relevant to the question asked, which is seeking a specific value (miles per gallon) for an average city bus. Here's a more accurate and relevant approach:

1. **Relevance and Accuracy**: The image does not contain any information about the fuel efficiency of a city bus, which is the core requirement of the question. Therefore, the answer provided by Assistant is accurate in the sense that the image does not provide the necessary data, but it might be more accurate to state that the image does not provide the specific value asked for.

2. **Relevance**: The image does show a city street with various vehicles, which might be relevant to the context of the question, but it does not provide the specific data needed to answer the question about the bus's fuel efficiency.

3. **Accuracy**: The statement that the image does not provide the necessary information is accurate, but it might be more accurate to specify that the image does not provide the specific value (miles per gallon) asked for in the question.

Given the lack of specific data in the image, the most accurate and relevant answer would be to state that the image does not provide the necessary information to determine the miles per gallon of an average city bus. This would be a more precise and accurate response to the question asked.</think>

**Score: 2**

### Perception-Judge-Flex-7B

Assistant provided a relevant and accurate answer to the question asked, which was indeed to determine the miles per gallon (MPG) of an average city bus, based on the information available in the image. Here's a detailed breakdown:

1. **Relevance and Accuracy**: The answer is accurate in its assessment that the image does not provide any specific data about the fuel efficiency of city buses. The image shows a typical urban street scene with various vehicles, but it does not include any information relevant to the question asked. This is a relevant observation because the question specifically asked for a figure related to bus fuel efficiency, which is not directly shown or implied in the image.

2. **Accuracy in Context**: The answer correctly pointed out that the image does not include any data on bus fuel efficiency, which is a relevant detail. It also correctly noted that the image includes a variety of vehicles, such as cars and a bicycle, which might be relevant to a broader discussion on urban transportation, but these do not pertain to the specific question asked about bus efficiency.

3. **Relevance to Question**: The relevance of the answer to the question asked is clear. The question specifically asked for the miles per gallon of an average city bus, and the image did not provide any such data. Therefore, the answer is accurate in stating that the image does not provide the necessary information to answer the question.

Given these considerations, the answer provided by Assistant is both accurate and relevant to the context of the question asked.</think>

**Score: 4**

*Figure S12.* Additional qualitative example of score setting.

### Instruction

**Nobel Prize winners**

| Country | 1980s | 1990s |
|---------|-------|-------|
| Germany | 11 | 7 |
| Italy | 3 | 1 |
| Canada | 3 | 8 |

This is a figure including mathematical information. Please analyze this figure in detail and answer the following question based on this figure: For an assignment, Johnny looked at which countries got the most Nobel Prizes in various decades. **In the 1990s, how many more Nobel Prize winners did Canada have than Italy? (Unit: Nobel Prize winners)**

### Response Model: LLaVA-1.5-13B

In the 1990s, Canada had 7 Nobel Prize winners, while Italy had 3. Therefore, Canada had 4 more Nobel Prize winners than Italy in the 1990s.

### Human

Score: 1 / 5

### Qwen2.5-VL-7B

The assistant's response is relevant and accurate. It correctly identifies the number of Nobel Prize winners in Canada and Italy for the 1990s and calculates the difference accurately. The response is clear and directly addresses the question without any extraneous information. The granularity is appropriate as it focuses on the specific decade and the difference in the number of winners between the two countries.

Judgement: 4

### Flex-Judge-VL-7B

Assistant's answer is accurate and relevant to the question asked. Here's a detailed breakdown:

1. **Relevance and Accuracy**: The answer correctly identifies the number of Nobel Prize winners in the 1990s for both Canada and Italy. According to the table, Canada had 8 Nobel Prize winners in the 1990s, and Italy had 1. The difference between these numbers is indeed 8 - 1 = 7, which means Canada had 7 more Nobel Prize winners than Italy in the 1990s. This is accurate and directly relevant to the question asked.

2. **Calculation**: The calculation provided by Assistant is correct. The difference between the number of Nobel Prize winners in the 1990s for Canada (8) and Italy (1) is 8 - 1 = 7. This means Canada had 7 more Nobel Prize winners than Italy in the 1990s, which aligns with the data in the table.

3. **Clarity and Precision**: The answer is clear and precise in its statement, "In the 1990s, Canada had 7 Nobel Prize winners, while Italy had 3. Therefore, Canada had 4 more Nobel Prize winners than Italy in the 1990s." This statement is accurate and directly answers the question asked, which is to find the difference in the number of Nobel Prize winners between Canada and Italy in the 1990s.

Given these points, the accuracy and relevance of the answer are quite high, making it a correct response to the question.</think>

**Score: 5**

### Perception-Judge-Flex-7B

Assistant's answer is inaccurate and somewhat misleading. Here's a detailed breakdown of the error and the correct approach:

1. **Incorrect Calculation**: The statement İn the 1990s, Canada had 7 Nobel Prize winners, while Italy had 3. Therefore, Canada had 4 more Nobel Prize winners than Italy in the 1990sïs incorrect. According to the table, in the 1990s, Canada actually had 8 Nobel Prize winners, while Italy had 1. The correct difference would be 8 (Canada) - 1 (Italy) = 7, not 4.

2. **Misinterpretation of the Data**: The table shows that in the 1990s, Canada had 8 Nobel Prize winners, which is more than the 1 that Italy had. Therefore, the correct answer should be that Canada had 7 more Nobel Prize winners than Italy in the 1990s.

To correct the answer, the correct statement would be: İn the 1990s, Canada had 8 Nobel Prize winners, while Italy had 1. Therefore, Canada had 7 more Nobel Prize winners than Italy in the 1990s.</think>

**Score: 1**

*Figure S13.* Additional qualitative example of score setting.

**Instruction**

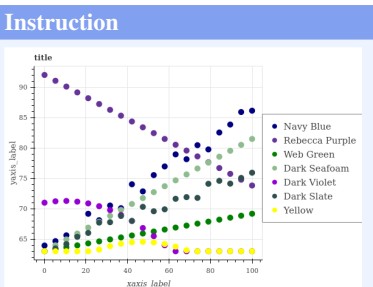

This is a figure including mathematical information. Please analyze this figure in detail and answer the following question based on this figure: Is Web Green greater than Yellow?

**Response Model: Gemini-1.0-Pro-Vision**

Web Green is not greater than Yellow. In the figure, the yellow line is above the web green line for all values of x-axis. Therefore, yellow is greater than web green.

**Human**

Score: 1 / 5

**Qwen2.5-VL-7B**

The assistant's response is relevant and accurate. It correctly identifies the relationship between the Web Green and Yellow lines in the figure, stating that Yellow is greater than Web Green for all values of the x-axis. The response is clear and concise, providing a straightforward answer to the question. However, the response lacks granularity as it does not delve into the specifics of the data or the exact points where the lines intersect or cross, which could have provided more detailed insight into the relationship between the two lines.

Judgement: 4

**Flex-Judge-VL-7B**

Assistant's answer is accurate and relevant to the question asked. Here's a detailed breakdown of why:

1. **Relevance and Accuracy**: The answer correctly identifies that the Yellow line is above the Web Green line throughout the entire range of the x-axis. This means that for any given value of x, the value of Yellow is greater than the value of Web Green. This is a direct interpretation of the graph, which is a relevant and accurate way to answer the question "Is Web Green greater than Yellow?"

2. **Level of Detail**: The answer is precise in its conclusion, specifying that the Yellow line is above the Web Green line for all values of x. This level of detail is accurate and relevant to the question, which asks whether Web Green is greater than Yellow. By stating that this is true for all x-values, the answer is comprehensive and accurate.

3. **Logical Reasoning**: The reasoning is logical and straightforward. It correctly identifies the relative positions of the lines on the graph, which is a clear and accurate way to determine the relationship between Web Green and Yellow.

In contrast, if the answer had been inaccurate, it might have suggested that Web Green is greater than Yellow at some points, which would be incorrect based on the graph. For instance, if the answer had stated that Web Green is greater than Yellow in some parts of the graph, but not in others, it would have been inaccurate because the graph clearly shows that Yellow is always above Web Green.

</think>

**Score: 5**

**Perception-Judge-Flex-7B**

Assistant's answer is inaccurate and somewhat misleading. Here's a detailed analysis to correct the issue:

1. **Accuracy and Relevance**: The accuracy of the answer can be questioned because the yellow line is indeed below the Web Green line for a significant portion of the x-axis range, not above it. This misinterpretation might lead to a misunderstanding of the relationship between the two lines.

2. **Correct Approach**: To determine whether Web Green is greater than Yellow, we need to compare the y-values of these two lines at each corresponding x-value. From the figure, it is clear that the Web Green line (green dots) is consistently above the Yellow line (yellow dots) for the entire x-axis range. This means Web Green is greater than Yellow for all values of x.

3. **Misinterpretation**: The statement "the yellow line is above the web green line for all values of x-axis" is incorrect. The correct interpretation should be that Web Green is always above the Yellow line, which would mean Web Green is greater than Yellow.

4. **Conclusion**: Based on the correct interpretation of the figure, Web Green is indeed greater than Yellow for all values of x. Therefore, the correct answer should be that Web Green is greater than Yellow, not the other way around.

</think>

**Score: 1**

*Figure S14.* Additional qualitative example of score setting.

### Instruction

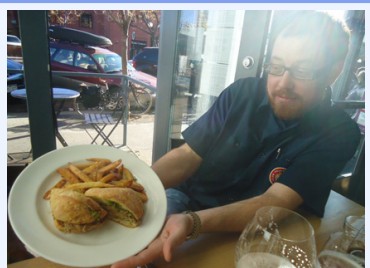

Please analyse this figure in detail and answer the following question based on this figure: **At what angle is the man holding the plate?**

### Response Model: CogVLM

The man is holding the plate at a 90-degree angle. This can be deduced by observing that the horizontal line of the plate is parallel to the bottom edge of the table, and the vertical line of the plate is perpendicular to the man's body, making it a 90-degree angle.

### Human

Score: 1 / 5

### Qwen2.5-VL-7B

The assistant's response is relevant and addresses the user's instruction by providing a specific angle at which the man is holding the plate. However, the explanation could be more detailed and precise. The assistant states that the plate is held at a 90-degree angle, but the reasoning provided is somewhat vague. A more detailed explanation, such as the alignment of the plate's edges with the table and the man's body, would enhance the comprehensiveness of the response.

Judgement: 4

### Flex-Judge-VL-7B

Assistant's answer is quite accurate in identifying the angle at which the man is holding the plate. Here's a detailed breakdown of why this is a correct analysis:

1. **Orientation of the Plate**: The plate is being held in a way that the horizontal line of the plate is parallel to the bottom edge of the table. This means the plate is being held horizontally, which is a 90-degree angle to the vertical plane of the man's body.

2. **Perpendicularity**: The vertical line of the plate is perpendicular to the man's body. This is a key detail that confirms the angle is 90 degrees, as the plate is being held in a way that its vertical edge is at a right angle to the man's body.

3. **Visual Confirmation**: The image shows the plate being held in a way that aligns with these descriptions. The man's hand is positioned such that the plate is horizontal, and the vertical edge of the plate is aligned with the vertical line of his body, which is consistent with a 90-degree angle.

Given these observations, the analysis provided by Assistant is accurate in identifying the angle at which the man is holding the plate.
</think>

**Score: 5**

### Perception-Judge-Flex-7B

Assistant's answer is quite accurate in identifying the angle at which the man is holding the plate. Here's a detailed breakdown of the reasoning:

1. **Angle Determination**: The man is holding the plate in such a way that the plate is perpendicular to the table, which means the angle between the plate and the table is 90 degrees. This is a correct observation.

2. **Observation of Lines**: The description mentions that the horizontal line of the plate is parallel to the bottom edge of the table, and the vertical line of the plate is perpendicular to the man's body. This is a clear and accurate way to describe the orientation of the plate, which is indeed a 90-degree angle.

3. **Contextual Accuracy**: The analysis is accurate in the context of the image, where the plate is being held in a way that it forms a right angle with the table. This is a reasonable interpretation based on the visual evidence in the image.

Given these accurate observations, the answer provided by Assistant is quite accurate and relevant to the question asked.
</think>

**Score: 4**

*Figure S15.* Representative failure cases of Perception Judge. The model struggles with fine-grained perceptual distinctions and scenes requiring implicit three-dimensional or complex spatial understanding, and occasionally fails on tasks that demand multi-step reasoning.

