# OpenReview forum: "Mitigating Perceptual Judgment Bias in Multimodal LLM-as-a-Judge via Perceptual Perturbation and Reward Modeling"
_ICML.cc/2026/Conference — ICML 2026 regular_

### Official Review · Reviewer_iTk4 · 2026-03-03

**Soundness:** 2
**Presentation:** 3
**Significance:** 2
**Originality:** 2
**Overall Recommendation:** 2
**Confidence:** 5

**Summary:**

This paper first identifies Perceptual Judgment Bias in MLLM judges, where MLLM judges tend to reward linguistically plausible narratives over perceptually correct answers for conflicts between visual evidence and textual responses. To address this issue, the authors curate a Perceptually Perturbed Judgment Dataset (PPJD) consisting of minimally edited counterfactual responses with manually injected perceptual and reasoning errors. Building on this dataset, they apply GRPO to fine-tune MLLM judge models, achieving improved performance on the MLLM-as-a-Judge benchmark.

**Compliance With Llm Reviewing Policy:**

Affirmed.

**Final Justification:**

While the rebuttal adds additional experiments and analyses, the core concerns are not fully resolved:

1) extending perturbation-based supervision for VLM reasoning into a judge-training pipeline is still not enough to establish a strong methodological contribution;
2) the training data curation pipeline remains largely restricted to verifiable prompts, which limits its practical scalability to open-ended settings where VLM-as-a-Judge is more useful.

In addition, the newly added Qwen2.5-VL results on VL-RewardBench appear unexpectedly low relative to prior work, which makes the comparison less convincing.

Therefore, I am still leaning toward rejection.

**Key Questions For Authors:**

- Why do the authors use multiplication instead of summation for the total reward? What is the rationale behind this design choice?

**Limitations:**

yes

**Strengths And Weaknesses:**

### Strengths

- The overall writing is clear and easy to follow.
- The paper formulates Perceptual Judgment Bias and distinguishes two failure modes (insufficient perceptual capability and response anchoring), which represent practical and important issues for multimodal judging systems.
- Performance gains are consistent across different base models and evaluation settings (score, pair, batch), validating the effectiveness of the proposed method.

### Weaknesses

- Injecting perceptual errors to golden responses has been explored in prior work for improving multimodal LLMs [1,2]. The technical novelty relative to prior work remains unclear. The authors should explicitly clarify their unique contributions and how their approach differs from existing methods.

  [1] Zhou, Yiyang, et al. "Aligning modalities in vision large language models via preference fine-tuning." *arXiv preprint arXiv:2402.11411* (2024).

  [2] Wang, Xiyao, et al. "Vicrit: A verifiable reinforcement learning proxy task for visual perception in vlms." *arXiv preprint arXiv:2506.10128* (2025).

- The constructed dataset is based on prompts with verifiable answers (e.g., multiple-choice or closed-form questions). In such settings, answer accuracy can be directly evaluated without requiring a judge model. As MLLM judges are particularly valuable in open-ended evaluation scenarios, it remains unclear whether the proposed method would yield comparable gains when judging open-ended responses without short final answers.

- Although the paper identifies two distinct failure modes, it lacks quantitative analysis disentangling their relative impact.

  - Which failure mode is more dominant for existing judge models?
  - To what extent does the proposed dataset mitigate each failure mode?

- Limited evaluation scope. Results are only reported on the MLLM-as-a-Judge Benchmark. It would be more convincing to report on additional reward benchmarks such as VL-RewardBench and Multimodal-RewardBench, and compare with more recent RL-finetuned MLLM judges such as [3-5].

  [3] UnifiedReward-Think: Wang, Yibin, et al. "Unified multimodal chain-of-thought reward model through reinforcement fine-tuning." *arXiv preprint arXiv:2505.03318* (2025).

  [4] R1-Reward: Zhang, Yi-Fan, et al. "R1-reward: Training multimodal reward model through stable reinforcement learning." *arXiv preprint arXiv:2505.02835* (2025).

  [5] Wang, Xiyao, et al. "Llava-critic-r1: Your critic model is secretly a strong policy model." *arXiv preprint arXiv:2509.00676* (2025).

---

> ### Author Rebuttal · Authors · 2026-03-31
>
> ## W1
> We thank the reviewer for this important point.
> We agree that perceptual perturbation has been used in prior work for improving VQA/VLM answer generation [1,2].
> Our novelty is not the perturbation primitive itself, but its **judge-specific use**: note that ordering supervision directly aligns with the judge's task, whose output is inherently a ranking over responses.
>
> This distinction matters because our analysis in W3 shows that the main failure mode of existing judge models is response anchoring, not simply weak perception: the model may answer the underlying VQA correctly, yet still over-reward a perceptually incorrect response. This suggests that improving VQA ability alone is not sufficient to improve judging.
>
> To verify this, we compare two uses of PPJD under the same backbone and training recipe. VLM DPO uses PPJD as standard answer-generation preference data, encouraging preference for the correct response itself. Judge DPO instead uses PPJD as evaluator supervision, encouraging the model to rank the correct response above perturbed ones. As shown below, VLM DPO yields little gain on judge benchmarks, whereas Judge DPO yields clear improvements. This shows that the benefit comes not from perceptual perturbation alone, but from turning it into evaluator-specific supervision that directly mitigates perceptual judgment bias.
>
> | Method | Score (↑) | Pair w. Tie (↑) | Pair w.o. Tie (↑) | Batch (↓) |
> |-|-:|-:|-:|-:|
> | Flex-Judge | 0.404 | 0.514 | 0.623 | 0.517 |
> | + VLM DPO | 0.418 | 0.509 | 0.610 | 0.518 |
> | + Judge DPO | 0.471 | 0.522 | 0.628 | 0.507 |
>
> ## W2
> We agree that MLLM judges are especially valuable in **open-ended evaluation**, and we will clarify this more explicitly in the paper.
>
> Importantly, our evaluation already includes many open-ended tasks where the goal is to judge response quality rather than verify a short final answer, including image description (CC-3M), image-quality explanation (Aes), visual reasoning (COCO), detailed image analysis (C.C.), creative generation (VisIT), and instruction-following outputs (M2W).
> Thus, our benchmark covers both directly verifiable-answer settings and genuinely open-ended judging scenarios.
> Our training signal teaches the judge to prefer correctly visually grounded responses over perceptually inconsistent ones, which is also the core ability required in open-ended evaluation.
> We agree and will highlight this motivation more prominently in our revision.
>
> ## W3
> We performed an additional quantitative analysis to disentangle the two failure modes.
> The results show that, in existing judge models, **response-anchoring** is the more dominant failure mode (14.0%) than insufficient perception (9.4%). Training with PPJD reduces both, but yields a notable gain on response anchoring (14.0% → 7.6%) than on insufficient perception (9.4% → 6.7%).
> Due to the space limit, we refer the reviewer to **Reviewer L9Vt W3** for the full quantitative breakdown and discussion in a controlled, verifiable perturbation setting.
>
> ## W4
> Our initial evaluation followed the standard MLLM-as-a-Judge protocol, using the MLLM-as-a-Judge benchmark under score, pairwise, and batch-ranking settings across 14 tasks, since this most directly matches our goal of improving **multimodal judgment reliability**.
>
> That said, we agree that a broader evaluation would strengthen the paper. Following the reviewer’s suggestion, we extended our experiments to **VL-RewardBench** and **Multimodal-RewardBench**, while also comparing with recent RL-finetuned reward/judge models such as **UnifiedReward-Think**, **R1-Reward**, and **LLaVA-Critic-R1**.
> These results show that the proposed method generalizes beyond the original benchmark, and we will include them in the revision.
>
> VL-RewardBench
> | -  | Gen. | Hall. | Reas. | Overall Acc. | Macro Acc. |
> |-|-:|-:|-:|-:|-:|
> | UnifiedReward-Think-7B | 78.1 | 72.7 | 66.0 | 73.8  | 72.3 |
> | R1-Reward-7B | 63.8 | 85.7 | 64.8 | 71.9  | 71.4 |
> | LLaVA-Critic-R1-7B | -  | -  | -  | 61.5  | -  |
> | Qwen3-VL-4B-Thinking | 50.4 | 78.3 | 76.4 | 73.6  | 68.4 |
> | Perception-Judge-Qwen3-4B | 53.8 | 79.0 | 78.8 | 75.2  | 70.5 |
>
> Multimodal-RewardBench
> | - | Overall | Corr. | Pref. | Know. | Math | Coding | Safety | VQA |
> |-|-:|-:|-:|-:|-:|-:|-:|-:|
> | UnifiedReward-Think-7B | 69.1 | 62.8 | 67.0 | 62.2 | 61.9 | 58.2 | 98.4 | 73.2 |
> | R1-Reward-7B | 82.2 | 77.5 | 74.0 | 74.9 | 83.1 | 79.6 | 99.6 | 86.5 |
> | LLaVA-Critic-R1-7B | 69.3 | 62.6 | 67.1 | 58.4 | 61.5 | 54.8 | 91.1 | 80.9 |
> | Qwen3-VL-4B-Thinking | 73.7 | 68.1 | 65.3 | 67.3 | 74.9 | 55.0 | 94.5 | 84.4 |
> | Perception-Judge-Qwen3-4B | 77.8 | 71.6 | 70.2 | 72.4 | 83.5 | 60.3 | 92.5 | 87.8 |
>
> MLLM-as-a-Judge Benchmark
> | - | Score (↑) | Pair w. Tie (↑) | Pair w.o. Tie (↑) | Batch (↓) |
> |-|-:|-:|-:|-:|
> | Perception-Judge-Qwen3-4B | 0.457 | 0.554 | 0.691 | 0.444 |
> | UnifiedReward-Think-7B | 0.257 | 0.459 | 0.552 | 0.591 |
> | R1-Reward-7B | 0.233 | 0.391 | 0.478 | 0.630 |
> | LLaVA-Critic-R1 | 0.350 | 0.383 | 0.412 | 0.587 |

---

> > ### Author Rebuttal · Reviewer_iTk4 · 2026-04-01
> >
> > Thanks to the authors for the additional clarifications and results. Some of my concerns remain unresolved:
> >
> > 1) It would be helpful to report results of Perception-Judge-Flex on VL-RewardBench and Multimodal RewardBench, since the compared models also use Qwen2.5-VL-7B as backbones, enabling a fair comparison.
> > 2) My main concern is still about technical novelty and insights. Applying perceptual perturbations within the judge framework alone appears insufficient to demonstrate a strong methodological contribution.
> > 3) Regarding the failure mode, while response anchoring is identified as the dominant issue, it is unclear why the Perceptually Perturbed dataset is the most appropriate solution. Could a simpler approach—such as prompting the judge to first answer the question and then provide a judgment—achieve similar improvements?

---

> > > ### Author Response · Authors · 2026-04-04
> > >
> > > We thank the reviewer for the thoughtful follow-up and are encouraged that our additional clarifications have at least partially addressed the concerns.
> > > We appreciate the opportunity to further clarify these points.
> > >
> > > ## R1
> > >
> > > We agree that evaluating Perception-Judge-Flex on VL-RewardBench and Multimodal RewardBench would provide a more direct comparison.
> > >
> > > We therefore conducted these **additional experiments**, and the results are provided below.
> > > While models based on Qwen2.5-VL-7B generally underperform those built on Qwen3-based backbones, the results show that our method consistently improves over the original base model on both benchmarks.
> > >
> > > VL-RewardBench
> > >
> > > | - | Gen. | Hall. | Reas. | Overall Acc. | Macro Acc. |
> > > |---|---:|---:|---:|---:|---:|
> > > | Qwen2.5-VL-7B-Instruct | 40.9 | 29.9 | 62.5 | 39.5 | 44.4 |
> > > | Flex-Judge-VL-7B | 35.6 | 40.7 | 59.7 | 44.9 | 45.4 |
> > > | Perception-Judge-Flex-7B | 44.3 | 55.4 | 56.1 | 54.0 | 52.0 |
> > >
> > > Multimodal RewardBench
> > >
> > > | - | Overall | Correctness | Preference | Knowledge | Math | Coding | Safety | VQA |
> > > |---|---:|---:|---:|---:|---:|---:|---:|---:|
> > > | Qwen2.5-VL-7B-Instruct | 56.0 | - | - | - | - | - | - | - |
> > > | Flex-Judge-VL-7B | 65.3 | 57.5 | 60.2 | 58.6 | 65.6 | 55.0 | 73.8 | 77.0 |
> > > | Perception-Judge-Flex-7B | 67.5 | 64.5 | 63.5 | 62.1 | 67.5 | 55.5 | 63.4 | 81.8 |
> > >
> > >
> > > ## R2
> > >
> > > We would like to clarify that our contribution is not limited to applying perceptual perturbations alone within an existing judge framework.
> > > While we are, to the best of our knowledge, the first to introduce such perturbations in the MLLM-as-a-Judge setting, our primary contribution lies in the analysis and methodology built upon this insight.
> > >
> > > One of our main contributions is to identify, formalize, and quantitatively analyze **Perceptual Judgment Bias** in the MLLM-as-a-Judge setting, including its two distinct failure modes: insufficient perceptual capability and response anchoring.
> > > In our view, this is fundamentally different from simply improving VQA or judge performance, as it addresses a distinct failure mode specific to the MLLM-as-a-Judge setting.
> > > In VQA, the model is asked to answer the question correctly; in MLLM-as-a-Judge, the model must evaluate another response while grounding its judgment in its own visual perception rather than relying on the response text.
> > > This creates a distinct failure mode that, to the best of our knowledge, has not been explicitly defined or systematically studied in prior work.
> > >
> > > Building on this analysis, we introduce PPJD, which is designed to mitigate both perceptual and reasoning errors through controlled perturbations, and combine it with a verifiable GRPO-based batch-ranking objective for perception-grounded judge alignment.
> > >
> > >
> > > ## R3
> > >
> > > We agree that prompting-based alternatives, such as asking the judge to first answer the question and then provide a judgment, are worth considering.
> > > However, we find that such approaches mainly improve reasoning format and do not resolve the underlying bias.
> > >
> > > To verify this, we evaluate an **answer-first prompting** variant of the judge.
> > > As shown in the table below, it yields only marginal improvements over Flex-Judge (76.55% → 77.09%), with only slight reductions in insufficient perception (9.40% → 9.16%) and response anchoring (14.05% → 13.75%).
> > > This indicates that response anchoring can persist even when the model is prompted to explicitly reason about the image, as the final judgment may still rely on the candidate response text.
> > >
> > > In contrast, our method provides explicit, verifiable supervision that directly trains the judge to penalize perceptual inconsistency.
> > > We therefore view prompting or CoT-style approaches as complementary to our method, rather than as a sufficient solution, as they encourage structured reasoning but do not guarantee perceptually grounded judgments.
> > >
> > > [**Bias Mitigation Results**]
> > >
> > > | Method | Overall Acc. ↑ | Total Error ↓ | Mode (1): Insufficient Perception ↓ | Mode (2): Response Anchoring ↓ |
> > > | :- | :-: | :-: | :-: | :-: |
> > > | Flex-Judge | 76.55% | 23.45% | 9.40% | 14.05% |
> > > |  + answer first prompt | 77.09% | 22.91% | 9.16% | 13.75% |
> > > |  + Our method | **85.71%** | **14.29%** | **6.74%** | **7.55%** |
> > >
> > > [**MLLM-as-a-Judge Benchmark**]
> > >
> > > | Method | Score↑ | Pair w. Tie ↑ | Pair w.o Tie ↑ | Batch ↓ |
> > > | :- | :-: | :-: | :-: | :-: |
> > > | Flex-Judge | 0.404 | 0.514 | 0.623 | 0.517|
> > > |  + answer first prompt | 0.413 | 0.518 | 0.619 | 0.519|
> > > |  + Our method | **0.466** | **0.520** | **0.645** | **0.505** |

---

### Official Review · Reviewer_L9Vt · 2026-03-08

**Soundness:** 2
**Presentation:** 3
**Significance:** 3
**Originality:** 3
**Overall Recommendation:** 4
**Confidence:** 5

**Summary:**

Judge models are easily impacted by the responses from the MLLMs, i.e., wrong answers but correct reasoning paths. To this end, this paper designs a training framework that constructs perceptual-perturbation preference data, and, based on it, proposes a model with RL and ranking-based reward functions to improve MLLM judges. Experiments show that the pipeline mitigates judgment bias and leads to more reliable evaluation.

**Compliance With Llm Reviewing Policy:**

Affirmed.

**Final Justification:**

Good to see the rebuttal. The rating is kept.

**Key Questions For Authors:**

See Major Weaknesses. The reviewer's current rating of \<Weak accept\> is contingent on a satisfactory rebuttal.

**Limitations:**

This paper contains sections discussing limitations of the paper, while, a statement on societal impact is missing.

This doesn't influence the reviewer's judgement and rating of the paper. However, as per the conference guidance, the statement shall be included. The reviewer suggests that authors take care of these.

**Strengths And Weaknesses:**

[Strengths]
1. The rationale behind this paper is good. The judge LLMs will be impacted by the reasoning paths and thus make wrong judgments. Efforts to confront this problem are applaudable.
2. The designed framework could serve as a reference for this community to further advance.
3. Extensive experiments are made to showcase the feasibility.
4. The presentations are good, and the paper is in a smooth flow to read.

[Major Weaknesses]
1. While the designed framework is promising, the outcomes are with some concerns. From the Tab. 1, especially for w. Tie and Batch, the gap is not that obvious against the baseline models. (0.514 v.s. 0.520). The concerns are further related to the ranking reward. Currently, the reward is too intuitive for readers. Any ablations upon the ranking gap choice, or making it more continuous for the rewards?
2. The automatic verification of the constructed data to filter unsuccessful perturbations is still unclear, will the generated $r_{rp}$ truly change only perceptual attributes without introducing unintended reasoning or artifacts? Any in-the-loop intervention from a human check?
3. Speaking of this, while, the paper is about bias, but the evaluation is made upon common datasets. Not sure if the capabilities of dealing with bias really improve.

[Minor Weaknesses]
1. We implementPerception - space missing there
2. For the GRPO training, the importance sampling is upon sequences or tokens?

---

> ### Author Rebuttal · Authors · 2026-03-31
>
> ## W1
> We thank the reviewer for this important point.
> We agree that the ranking reward design can be better justified.
>
> To address this, we conducted additional reward ablations using alternative continuous ranking objectives beyond our current Levenshtein-based reward.
> Specifically, we compared: (i) our Levenshtein-based ranking reward, (ii) a Kendall’s tau-based reward, and (iii) an NDCG-based reward.
> All three reward designs improved over the Flex-Judge (NeurIPS 2025) baseline, which suggests that the overall benefit is not tied to a single hand-crafted gap choice.
> At the same time, the Levenshtein-based reward gave the best overall performance.
>
> This result indicates that the method is reasonably robust across different ranking-gap formulations, while the Levenshtein version provides the strongest empirical trade-off in our setting.
> We will add this ablation and clarify the rationale for the reward choice in the revision.
>
> Additionally, the small gap on Pair w. Tie accuracy does not fully reflect changes in ranking quality.
> The improvement is also visible on supplementary results (Table S3.), which further show consistent gains on complementary ranking metrics, including Spearman, F1, Recall, Kendall’s tau, and NDCG.
> This supports that the reward improves ranking behavior more broadly, rather than optimizing a single discrete metric.
>
> |Type|S (↑)|P w. Tie (↑)|P w.o. Tie (↑)|B (↓)|
> |-|-:|-:|-:|-:|
> | Flex-Judge | 0.404 | 0.514 | 0.623 | 0.517 |
> | + Levenshtein | **0.466** | **0.520** | **0.645** | **0.505** |
> | + Kendall's Tau | 0.447 | 0.514 | 0.634 | 0.509 |
> | + NDCG | 0.455 | 0.517 | 0.640 | 0.510 |
>
> ## W2
> We agree that the description of the automatic verification step was not sufficiently clear.
>
> As described in the last part of the Detailed Dataset Generation Pipeline in Supp. D, the automatic filtering stage deterministically checks whether the final answer of the generated rejected response differs from that of the original correct response.
> More specifically, starting from a correct (chosen) response, we generate a perturbed rejected response and then verify whether its deterministically verifiable answer is actually changed to a wrong answer.
> If the perturbed response has the same answer as the original correct response, we regard that perturbation as unsuccessful and exclude it from the dataset.
>
> This filtering step is fully deterministic and does not involve human-in-the-loop intervention.
>
> Separately, during development, we used a small validation subset of the training data (about 5%) to inspect perturbation results and refine the prompting pipeline.
> We agree that, at scale, it is difficult to guarantee that every example is completely free of unintended reasoning changes or artifacts
> Our claim is therefore not that the dataset is perfectly noise-free, but that this construction provides a scalable and effective way to produce useful supervision in practice, as reflected in the final performance gains.
>
> We appreciate this comment and will revise the paper to clarify this part more explicitly.
>
> ## W3
> We agree that improvements on common benchmarks alone do not directly establish that the model better mitigates Perceptual Judgment Bias.
> For this reason, we also evaluate in a controlled and verifiable perturbation setting, where the bias can be probed more directly. As shown in Figure 1, our method consistently improves the judge’s ability to distinguish perceptually perturbed responses, reducing the overall perturbation error from 23.45% to 14.29%. This provides direct evidence of improved robustness to Perceptual Judgment Bias beyond aggregate benchmark gains.
> To better understand where this improvement comes from, we further analyze two failure modes on a verifiable short-answer VQA subset.
>
> Failure mode (1), **insufficient perceptual capability**, refers to cases where the judge cannot answer the underlying VQA correctly and also fails to detect the perceptual error.
> Failure mode (2), **response anchoring**, refers to cases where the judge answers the VQA correctly, but still fails to penalize a perceptually incorrect answer.
> We find that response anchoring is more dominant in existing judge models (14.0% vs. 9.4%).
> PPJD-based training reduces both modes, with a larger reduction on response anchoring (14.0% → 7.6%) than on insufficient perception (9.4% → 6.7%).
> Overall, total perturbation error decreases from 23.5% to 14.3%.
> We will add this breakdown in the revision.
>
> | Model | Overall Acc. ↑ | Total Error ↓ | Mode (1): Insufficient Perception ↓ | Mode (2): Response Anchoring ↓ |
> |-|-:|-:|-:|-:|
> | Flex-Judge | 76.55% | 23.45% | 9.40% | 14.02% |
> | + Ours | 85.71% | 14.29% | 6.74% | 7.55% |
>
> ## MW1
> We use token-level importance sampling, and we will clarify this in the revised version.
>
> ## L1
> We acknowledge that the current manuscript includes a discussion of limitations but does not yet provide a statement on societal impact.
> We will add an appropriate statement in the revised version.

---

> > ### Author Rebuttal · Reviewer_L9Vt · 2026-04-03
> >
> > Thanks for the rebuttal. I keep my score.

---

### Official Review · Reviewer_iqX2 · 2026-03-09

**Soundness:** 3
**Presentation:** 3
**Significance:** 3
**Originality:** 4
**Overall Recommendation:** 4
**Confidence:** 2

**Summary:**

This paper introduces an MLLM evaluation dataset called the Perceptually Perturbed Judgment Dataset to mitigate the issue of inaccurate results caused by visual perception biases in MLLM evaluation. The authors categorize such issues into two failure modes: either the MLLM evaluator has insufficient visual perception capabilities, or it correctly recognizes the image content but is misled by contradictory text generated by the responding model. To address this, the authors propose perceptual perturbation and reasoning perturbation datasets, and train the model using GRPO combined with a batch ranking objective. Experimental results demonstrate that the proposed method achieves strong performance across various multimodal benchmarks and different models.

**Compliance With Llm Reviewing Policy:**

Affirmed.

**Final Justification:**

The rebuttal adequately clarifies the definitions and design choices around perception–reasoning disentanglement and evaluation protocol. So I maintain my final recommendation.

**Key Questions For Authors:**

- Is it possible that evaluator MLLMs sometimes use the responding model's visual descriptions to correct their own judgments, rather than just being misled? If so, why does the paper treat this reliance as predominantly negative?
- Can the data generation process reliably distinguish perception and reasoning in highly coupled scenarios (e.g., complex chart interpretation)?
- Would the method fail or perform poorly when evaluating two candidate responses that contain distinct, yet equally severe perceptual errors?
- Could you clarify how the tripling of the data impacts training efficiency?

**Limitations:**

yes

**Strengths And Weaknesses:**

Strengths:

- The paper studies automated multimodal evaluation, which is becoming increasingly important as the community relies more on LLM-based judges.
- The method separates perception errors from reasoning errors and creates two types of perturbed data. This design helps expose weaknesses in the perception and reasoning abilities of current MLLM evaluators.
- The authors introduce the Perceptually Perturbed Judgment Dataset and the Batch Ranking Reward. These are interesting ideas that aim to improve the reliability of the LLM-as-a-Judge framework.
- The approach is evaluated on several benchmarks and multiple models, which helps demonstrate the robustness of the method.

Weaknesses:
- It is not clear to me that how perception and reasoning errors are reliably separated when generating the perturbed data, especially for tasks where the two are tightly connected.
- The paper assumes that the responding model’s visual description always harms the evaluator. However, in some cases such descriptions may actually help the evaluator correct its judgment. This possibility is not discussed.
- The paper does not explain how the method handles situations where both candidate answers contain different but equally serious perception errors.

---

> ### Author Rebuttal · Authors · 2026-03-31
>
> ## W1
>
> We thank the reviewer for this important question.
> We agree that perception and reasoning can be tightly coupled, so a perfectly clean disentanglement is difficult in some cases.
>
> In our construction, we define a perception error as an error that arises when visual information is translated into text, and we define a reasoning error as an error that occurs after that perceived information is fixed.
> Based on this definition, $r_{r_p}$ is constructed to mainly alter visually grounded attributes while keeping the subsequent reasoning as consistent as possible, and $r_{r_{p+r}}$ is then constructed by additionally introducing reasoning errors.
>
> In other words, our goal is not to claim perfect disentanglement in every example, but to provide a controlled and practical separation of the dominant error source.
> We will clarify this definition and limitation more explicitly in the paper.
>
>
> ## W2
>
> We thank the reviewer for this valuable point.
> We agree that, in some cases, the responding model’s visual description may help the evaluator correct its judgment.
>
> Our point is not that such descriptions are inherently harmful.
> Rather, the problem arises when the evaluator relies on them without sufficiently verifying them against the visual evidence.
> In that case, helpful descriptions can improve judgment, but unsupported or incorrect descriptions can also mislead it.
>
> Our goal is therefore not to discourage the use of response-provided visual descriptions, but to make the judge evaluate them more critically and ground its decision in its own perception.
> We will clarify this point more explicitly in the paper.
>
>
> ## W3
>
> In the pairwise evaluation setting, if both candidate responses contain different but equally severe perceptual errors, the ideal behavior is to assign low scores to both responses and treat the comparison as a tie, rather than force an arbitrary preference.
> More generally, our method is designed to penalize perceptual inconsistency in each response independently, not to assume that one response must always be better.
>
> In our PPJD construction, however, this ambiguity does not arise within the intended supervision pairs.
> Specifically, $r_{rp}$ contains a perceptual error, whereas $r_{rp+r}$ additionally includes a reasoning error, so the target ordering remains well-defined as $r_c > r_{rp} > r_{rp+r}$.
> We will clarify in the paper that ties are the appropriate outcome in evaluation when two responses exhibit comparably severe perceptual errors, while our training data is constructed to avoid such ambiguity.
>
>
>
> ## Q1
>
>
> We thank the reviewer for these thoughtful questions.
>
> Q1. We address this point in our response to Weakness 2 (W2).
>
> Q2. We address this point in our response to Weakness 1 (W1).
>
> Q3. We address this point in our response to Weakness 3 (W3).
>
> Q4. We address this point in our response to **Reviewer Z1s2 W1**, where we provide additional discussion on the 3k vs. 10k data setting and clarify the efficiency-performance trade-off of the expanded training tuples.

---

> > ### Author Rebuttal · Reviewer_iqX2 · 2026-04-03
> >
> > Thank you for your detailed response to my concerns and questions. I will keep my score.

---

### Official Review · Reviewer_Z1s2 · 2026-03-11

**Soundness:** 3
**Presentation:** 3
**Significance:** 3
**Originality:** 3
**Overall Recommendation:** 4
**Confidence:** 3

**Summary:**

This paper studies an important failure mode of multimodal judges: when visual evidence conflicts with textual descriptions, existing MLLM-as-a-Judge systems often prefer responses that are linguistically plausible but perceptually incorrect. This observation is consistent with prior work showing that multimodal judges suffer from bias, hallucination, and inconsistency. The paper formalizes this phenomenon as Perceptual Judgment Bias and proposes a two-part solution: (1) constructing the PPJD dataset with both perceptual perturbations and reasoning perturbations, and (2) training the judge with GRPO and a batch-ranking reward to improve perceptual fidelity, ranking consistency, and alignment with human evaluation.

**Compliance With Llm Reviewing Policy:**

Affirmed.

**Final Justification:**

The rebuttal has addressed my concerns, and I will maintain my original score.

**Key Questions For Authors:**

N/A

**Limitations:**

yes

**Strengths And Weaknesses:**

Strengths
1.  The paper identifies a meaningful and practically relevant failure mode of multimodal judges, namely their tendency to prioritize textual plausibility over visual correctness. This problem is well aligned with limitations discussed in prior MLLM-as-a-Judge literature and has clear research significance.
2. The work focuses specifically on judgment bias under visual-text conflict and introduces verifiable supervision through minimally edited counterfactual perturbations. This framing is relatively novel and provides a useful perspective beyond standard judge bias analyses.
3. The experimental evaluation is fairly broad. The paper reports results under score, pairwise, and batch-ranking protocols across 14 benchmark tasks, and also includes ablations and cross-backbone validation.

Weaknesses
1. The strongest results are mainly established on a relatively small 3k PPJD dataset and two backbone families. The paper lacks more systematic analysis on data scaling, perturbation strength, and unified comparisons against a wider set of recent strong backbones and judge models.
2. The dataset construction pipeline relies on proprietary models (e.g., GPT-series models) to generate and verify supervision signals. This limits full external reproducibility and may also introduce biases from closed-source models into the training distribution.
3. The method uses GRPO with multiple rollouts, but the paper does not report training cost or inference overhead. This makes it difficult to assess the practical usability and efficiency of the proposed approach.
4. Some key details of the PPJD construction process are under-specified, such as the retention rate after automatic filtering and the results of manual quality checks. As a result, it is difficult to judge the quality of the supervision signal and whether the dataset may contain systematic noise.

---

> ### Author Rebuttal · Authors · 2026-03-31
>
> ## W1
>
>
> We thank the reviewer for this helpful suggestion.
>
> For **scaling**, we additionally evaluated a 10k version of PPJD in Table 4.
> In Table 1, our 3k setting is hard samples, a quality-controlled subset aimed at maximizing data efficiency through verifiable supervision; the added comparison shows that 3k already gives strong gains, while 10k can further improve performance, clarifying the efficiency–performance trade-off.
>
> | Method | Score↑ | Pair w. Tie ↑ | Pair w.o Tie ↑ | Batch ↓ |
> |---|---:|---:|---:|---:|
> | Flex-Judge | 0.404 | 0.514 | 0.623 | 0.517 |
> | + PPJD 3k | 0.466 | 0.520 | 0.645 | 0.505 |
> | + PPJD 10k | 0.476 | 0.518 | 0.648 | 0.500 |
>
>
> For **perturbation strength**, our design includes a graded difficulty axis through $r_{r_p}$ and $r_{r_{p+r}}$, which is directly used in the ranking objective.
> We agree, however, that this was not framed clearly enough as a perturbation-strength analysis, and we will make this more explicit in the revision.
>
>
> For **broader model comparisons**, we also agree that recent strong judge and reward-model backbones should be included more systematically.
> In addition to the larger-scale backbones already reported in the supplementary material (Table S2; e.g., Flex-Judge-VL-32B, Qwen3-VL-8B-Thinking, Perception-Judge-Flex-32B, and Perception-Judge-Qwen3-8B), we added unified comparisons against recent reward models such as UnifiedReward-Think, R1-Reward, and LLaVA-Critic-R1 in the final results table.
> These additional comparisons further support that the benefit of our method is not limited to a single backbone family, but transfers across stronger recent multimodal judge/reward models as well.
>
> | Method | Score (↑) | Pair w. Tie (↑) | Pair w.o. Tie (↑) | Batch (↓) |
> |---|---:|---:|---:|---:|
> | UnifiedReward-Think-7B | 0.257 | 0.459 | 0.552 | 0.591 |
> | R1-Reward-7B | 0.233 | 0.391 | 0.478 | 0.630 |
> | LLaVA-Critic-R1-7B | 0.350 | 0.383 | 0.412 | 0.587 |
> | Perception-Judge-Qwen3-4B | 0.457 | 0.554 | 0.691 | 0.444 |
>
>
>
>
> ## W2
>
> We thank the reviewer for this important point.
> We agree that using proprietary models limits full reproducibility and may introduce bias.
> That said, our pipeline is **not an unconstrained end-to-end** generation process: we decompose it into controlled stages with human-designed constraints, and apply deterministic verification to discard samples whose perturbations do not produce the intended answer change.
> We also used a small validation split (~5%) to refine stage-specific prompts and improve consistency.
> While this does not fully remove the limitation, we believe it substantially mitigates the issue relative to end-to-end generation, and we will clarify this point in the revision.
>
>
>
> ## W3
>
> We thank the reviewer for this important point.
> We agree that computational cost and practical usability should be stated more clearly.
> GRPO with multiple rollouts (5) is used only during training, and the associated overhead is mitigated in practice by the optimized verl framework (e.g., 2 hours for Perception-Judge-Flex).
> At inference time, our method uses the same final judge model with no additional rollout or test-time tuning, so it does not introduce extra deployment-time overhead.
> We will also clarify that the 3-sample averaging in Table 1 is used only for evaluation stability, not as a required inference setting.
> While some implementation details are already included in the supplementary material (Sec. C), we will make this distinction more explicit in the revision.
>
>
> ## W4
>
> We thank the reviewer for this important point.
> We will clarify this more explicitly in the revision.
> In the filtering stage that compares each generated result against the ground-truth answer, approximately 63.2% of the generated samples are retained and used to construct the original 10k training pool.
> We will report this retention statistic more clearly to make the quality of the supervision signal more transparent.

---

> > ### Author Rebuttal · Reviewer_Z1s2 · 2026-04-03
> >
> > Thank you for the rebuttal. I will maintain my score.

---

### Decision · Program_Chairs · 2026-04-30

**Decision:**

Accept (regular)

**Comment:**

This paper studies a failure mode in multimodal LLM-as-a-Judge systems, where models favor plausible text over perceptually correct answers, and proposes a method based on perceptual perturbation data and reward-based ranking to encourage perceptually grounded judgments.

Reviewers broadly agree that the paper addresses an important issue and provides solid empirical validation. Three reviewers support acceptance and maintained their scores after rebuttal.

The main concern of iTk4 is novelty, as the approach is closely related to prior perturbation-based supervision (e.g., ViCrit) and reward modeling. The additional experiments strengthen the empirical case (including broader benchmarks and comparisons to simple baselines), but do not fully resolve this concern.

That said, the work distinguishes itself by targeting a judge-specific failure mode and shifting from perceptual error detection to ranking-based supervision to mitigate bias, making the contribution empirically convincing.